

# Multi-decadal trends and variability in burned area from the 5th version of the Global Fire Emissions Database (GFED5)

Yang Chen[1], Joanne Hall[2], Dave van Wees[3], Niels Andela[4], Stijn Hantson[5], Louis Giglio[2], Guido R. van der Werf[3], Douglas C. Morton[6], James T. Randerson[1]

[1]Department of Earth System Science, University of California, Irvine, CA, USA
[2]Department of Geographical Sciences, University of Maryland, College Park, MD, USA
[3]Faculty of Science, Vrije Universiteit Amsterdam, Amsterdam, Netherlands
[4]BeZero Carbon, London, UK
[5]Earth System Science Program, Faculty of Natural Sciences, Universidad del Rosario, Bogota, Colombia
[6]Biospheric Sciences Laboratory, NASA Goddard Space Flight Center, Greenbelt, MD, USA

*Correspondence to*: Yang Chen (yang.chen@uci.edu)

**Abstract.** Long-term records of burned area are needed to understand wildfire dynamics, assess fire impacts on ecosystems and air quality, and improve fire forecasts. Here we fuse multiple streams of remote sensing data to create a 24-year (1997-2020) dataset of monthly burned area as a component of the 5th version of the Global Fire Emissions Database (GFED5).

During 2001-2020, we use the Moderate Resolution Imaging Spectroradiometer (MODIS) MCD64A1 burned area product and correct for the errors of commission and omission. Adjustment factors are estimated based on region, land cover, and tree cover fraction, using spatiotemporally aligned burned area from Landsat or Sentinel-2. Burned area in cropland, peatland, and deforestation regions is estimated from MODIS active fire detections. Along Track Scanning Radiometer (ATSR) and Visible and Infrared Scanner (VIRS) active fire data are used to extend the time series back to 1997. Global annual burned area is

estimated to be 774±63 Mha yr$^{-1}$ during 2001-2020 or 5.9±0.5% of ice-free land. Burned area declined by 1.21±0.66% yr$^{-1}$, a cumulative decrease of 24.2±13.2% over 20 years. The global reduction is primarily driven by decreases in fire within savannas, grasslands, and croplands. Forest, peat, and deforestation fires did not exhibit long-term trends. GFED5 global burned area is 93% higher than MCD64A1, 61% higher than GFED4s, and in closer agreement with burned area products from higher-resolution satellite sensors. These data may reduce discrepancies between fire emission estimates from top-down

and bottom-up approaches, and improve understanding of global fire impacts on the carbon cycle and climate system.

## 1 Introduction

Wildfire is an integral part of the Earth system (Bowman et al., 2009), influencing the structure and functioning of many terrestrial ecosystems (Beerling and Osborne, 2006; Bond, 2016). Humans have a long history of using fire to enhance ecosystem services (Bowman et al., 2011), including the use of prescribed fire to maintain habitat for key animal and plant

species, increase crop productivity, and control plant diseases (Pyne, 2020). Fires on Earth exhibit substantial temporal and spatial variability on regional to global scales as a consequence of both natural variability and human influence. In particular, changes in factors such as fire occurrence, fuel amount, fuel moisture, fuel continuity, fire spread, fire severity, and fire





suppression, resulting from multiple global change drivers, have significantly altered fire regimes in recent decades (Archibald et al., 2013; van der Werf et al., 2017).

The areal extent of burning is an essential fire characteristic that regulates the amount of gas and particulate matter released into the atmosphere (Seiler and Crutzen, 1980), and the scope of fire impacts on ecosystem function. Accurate delineation of burned area can improve our understanding of fire impacts on the global carbon cycle, atmospheric composition, and climate (van der Werf et al., 2017); fire predictions on daily, seasonal, interannual, and decadal timescales (Chen et al., 2016; Taylor et al., 2013); the development and validation of prognostic fire models (Pereira et al., 2022); and the identification of important

climate, ecosystem, and human fire mechanisms that are responsible for variability and trends in fire dynamics (Andela et al., 2017). In this context, burned area mapping provides essential information for a broad community of science and management stakeholders interested in ecosystem conservation, climate mitigation, and approaches for limiting fire impacts on human health and infrastructure.

Before the satellite era, burned area was often estimated using field surveys, aerial photographs, and expert opinion, often with

statistics aggregated to regional, state, or country-wide areas by workers in fire management agencies (Crutzen and Andreae, 1990). Recent advances in satellite remote sensing have provided an alternative approach for mapping burned areas in a consistent and timely manner (Chuvieco et al., 2019; Giglio et al., 2009). Satellite-based detection has also considerably increased the spatial coverage and resolution of burned area mapping. Most satellite-based algorithms compare the spectral surface reflectance of post-fire images with those of pre-fire images to map burned areas, often using fire-specific vegetation

indices in the change detection algorithm (Alonso-Canas and Chuvieco, 2015; Eidenshink et al., 2007; Giglio et al., 2009; Roy et al., 2008). The quality of different satellite-derived burned area products, including the accuracy, uncertainty, frequency, coverage, and resolution, can vary considerably and is dependent on factors such as sensor design, orbit characteristics, and retrieval algorithms (Chuvieco et al., 2019).

At a global scale, near-daily repeat coverage from the Moderate Resolution Imaging Spectroradiometer (MODIS) instruments

onboard the Terra and Aqua satellites enabled important advances in burned area mapping. High-quality near infrared and shortwave infrared bands at a 500-m spatial resolution provided the basis for change detection using surface reflectance indices sensitive to the impact of fire on vegetation (Giglio et al., 2009; Roy et al., 2008). At the same time, analysis of MODIS mid-infrared and thermal infrared bands (and information from other wavelengths) provided a concurrent set of active fire observations (Giglio et al., 2016), which have been shown to be highly effective in providing training data needed to separate

burned and unburned areas in a locally-adaptive manner (Giglio et al., 2018). Analysis of these MODIS fire products has enabled the characterization of long-term burned area trends (Andela et al., 2017), regional variations in fire number and fire size (Andela et al., 2019), and the effects of climate on fire occurrence, expansion, and duration (Balch et al., 2022; Gutierrez et al., 2021).



More recently, a revolution is underway in estimating regional and global burned area using higher resolution satellite imagery
from Landsat (Hawbaker et al., 2017) and Sentinel-2 (Roteta et al., 2019). Several technical advances have made it practical
to map burned areas at continental scales at 20-m (Sentinel-2) or 30-m (Landsat) spatial resolution. First, with two Sentinel-2
satellites launched in 2015 and 2017 in complementary orbits, the revisit time for tropical areas is now less than 5 days (Drusch
et al., 2012). This more frequent coverage is essential for mapping fires in tropical areas where there is considerable cloud
cover and the imprint of a grassland fire on land surface reflectance may persist only for a period of several weeks, depending
on the timing of the fire relative to the onset of the wet season (Melchiorre and Boschetti, 2018). Second, improved
computation and data infrastructure, including cloud computing and the software infrastructure provided by the Google Earth
Engine and other online services, has made it easier to work with the large data volumes required for burned area mapping at
this resolution (Franquesa et al., 2020; Roteta et al., 2021). Third, a new generation of thermal anomaly and active fire
detection from the Visible and Infrared Imaging Spectroradiometer (VIIRS) on Suomi-NPP and NOAA-20 satellites at a 375-
m resolution (Schroeder et al., 2014) provides access to a higher-resolution and more sensitive set of fire detections for use in
the design and validation of burned area mapping algorithms. Compared to the coarser-resolution burned area products from
MODIS, the Landsat and Sentinel-2 burned area products are more effective in mapping burned areas from small fires.
Analysis of the Sentinel-2 burned area product from Africa has revealed an 80% increase relative to the standard MODIS-
based NASA burned area product (MCD64A1 C6) for the year 2016 (Ramo et al., 2021; Roteta et al., 2019).

However, within cropland - a quintessential small fire landscape - there are still technological limitations that impact the ability
to accurately map the full extent of burned area. These fires occur on a heterogeneous, managed landscape where the burn scar
is often manipulated by plowing or seeding within hours or days after the fire (Hall et al., 2021a). Therefore, the 3-5 day
combined Sentinel-2 and Landsat revisit time is often too long to capture all the burns. While MODIS data has more temporal
coverage, field sizes are typically smaller than 1-2 MODIS pixels, so the spectral reflectance change induced by fire is often
not strong enough to trigger thresholds in the burned area detection algorithm. This leads to an underestimation from MODIS-
based burned area products (Hall et al., 2016; Hall et al., 2021b). Consequently, active fire products are very useful for
identifying these small burns, although the timing and short duration of actively burning fires may impact the ability to map
the full extent of agricultural burns with the current generation of satellite sensors.

While the future of global burned area mapping is likely to occur at the higher resolution of 20-30 m, leveraging the combined
information provided by Sentinel-2 and Landsat (Claverie et al., 2018), the time series of more frequent observations required
for burned area mapping in many tropical areas is quite limited. There is a need to understand longer-term trends and variability
in burned area for many science and management applications. An important challenge in this regard is to find ways to
harmonize the information contained in the earlier (and coarser-resolution) fire products with the information provided by the
new Sentinel-2 and Landsat data, drawing upon the higher sensitivity to fire detection but shorter duration (and limited spatial
coverage) of the newer products.



In this study, we combine information from coarser (MODIS) and finer (Landsat and Sentinel-2) resolution satellite imagery time series to create a 24-year (1997-2020) record of global burned area at 0.25° spatial and monthly temporal resolution, as a component of the 5th version of the Global Fire Emissions Database (GFED5). The 20-year (2001-2020) MODIS active fire (Giglio et al., 2016) and burned area (Giglio et al., 2018) products serve as the backbone of the time series we develop. By comparing spatiotemporally aligned burned area estimates for selected reference scenes, we derive region- and vegetation-specific corrections for reducing omission and commission errors in the 500-m MODIS burned area. An explicit cropland-specific burned area methodology that relies on active fire detections is used to improve the quantification of crop-residue burned area (Hall et al., submitted). We also use active fire detections from the Along Track Scanning Radiometer (ATSR) (Arino et al., 1999) and the Visible and Infrared Scanner (VIRS) (Giglio et al., 2003) to extrapolate the adjusted burned area to include the pre-MODIS era (1997-2000), although the spatiotemporal variability is not as well preserved as during the MODIS era.

The details of the algorithms and data used for deriving the burned area are presented in section 2. In section 3, we report long-term trends and variability of regional and global burned area with our new time series. We also compare our data with multiple independent regional and global burned area products. In section 4, we discuss the implications and uncertainty of the new dataset and identify directions for future research. We describe our main conclusions in section 5.

## 2 Data and method

### 2.1 Method overview

Our approach to estimate global burned area (Fig. S1) during the MODIS era (2001-2020) takes advantage of the higher spatial resolution and detection sensitivity of burned area products from Landsat and Sentinel-2, as well as the higher temporal frequency, global coverage, and long time series of the Terra and Aqua MODIS fire products.

For most land cover types (denoted as *normal* cover types), we used the monthly MODIS MCD64A1 burned area product (Section 2.2) as the basis for estimating GFED5 burned area in each $0.25° \times 0.25°$ grid cell ($x$) and monthly time step ($t$), as shown by the $BA_{MCD64A1}$ term in Eq. (1). By comparing this dataset with a suite of reference burned area datasets derived from Landsat or Sentinel-2 imagery (Section 2.3), we made two adjustments to account separately for commission and omission errors at the coarser 0.25° resolution. Within MODIS burned perimeters, we expect the presence of sub-pixel unburned islands, considering that not all the area within a 500-m cell is required to burn in order for the MODIS burned area algorithm to flag a pixel as burned (Giglio et al., 2018). We adjusted for this commission error by multiplying the MCD64A1 burned area with a correction scalar ($\alpha$) derived from higher-resolution burned area and spatiotemporally aligned MCD64A1 burned area images, allowing $\alpha$ to vary as a function of vegetation type ($v$), tree cover fraction bin ($tc$), and continental-scale GFED region ($reg$). The scalar $\alpha$ is unitless (the ratio of two different areas) and typically varies from 0.5 to 0.9. The overall correction for commission errors using information from $\alpha$ is given by the first term in the right-hand side of Eq. (1), with 1 minus $\alpha$



representing the fraction of unburned islands within the MCD64A1 burned area according to the higher-resolution burned area data.

$$BA_{GFED5-norm}(x,t,v) = \sum_{tc}(BA_{MCD64A1}(x,t,v,tc) \times \alpha(v,tc,reg) + AF_{MODIS,out}(x,t,v,tc) \times \beta(v,tc,reg)) \qquad (1)$$

At the same time, there are also small fires or incompletely mapped burned areas by the MCD64A1 burned area algorithm, mostly due to its coarser spatial resolution. These omission errors have been shown to contribute to low biases in burned area at regional and global scales, especially at the beginning and end of the fire season (Ramo et al., 2021; Randerson et al., 2012). We corrected for omission errors by multiplying the surface area of MODIS active fire detections outside the perimeter of 500-m MCD64A1 burned area pixels ($AF_{MODIS,out}$) with a scalar, $\beta$, derived from the ratio of Landsat or Sentinel-2 reference

burned area to the corresponding $AF_{MODIS,out}$, again as a function of vegetation type, tree cover fraction bin, and GFED region. The scalar $\beta$ is also unitless (the ratio of two different areas) and typically varies between 0.5 and 4.0. The overall term describing the omission error correction using $\beta$ is given by the second term in the right-hand side of Eq. (1). The two sets of correction scalars $\alpha$ and $\beta$ used in Eq. (1) were derived using the aggregated sums of burned area (in each 0.25° grid cell) from MODIS and the higher-resolution data (Landsat or Sentinel-2) in different reference tiles (Table 1).

Eq. (1) was only used to estimate *normal* burned area ($BA_{GFED5-norm}$), which does not include vegetation burning in croplands ($BA_{GFED5-crop}$), tropical peatlands ($BA_{GFED5-peat}$), or that associated with deforestation ($BA_{GFED5-defo}$). Burned areas for these fire types are often difficult to map using MODIS, so we used specific approaches that rely on the scaling of active fire detections ($AF_{MODIS}$) to calculate the GFED5 burned area for these cases. For cropland burned area estimates, we used the Global Cropland Area Burned (GloCAB; Hall et al., submitted) product which scales MODIS active fire pixels to match

manually mapped crop-specific burned area reference data (Hall et al., 2021a) across major global agricultural regions. The scalar for peatland burning was derived by comparing the surface area of MODIS active fire detections with burned area mapped using Landsat imagery in the peatlands of Mawas, Central Kalimantan, Indonesia. Peat areas where the algorithm was applied both within Mawas and across the tropics as a whole were identified using version 2 of the Global Wetlands map (Gumbricht et al., 2017). The scalar for deforestation burning was derived by comparing the cumulative surface area of

MODIS active fire detections in deforestation areas with the long-term deforestation area reported by PRODES (Programa de Monitoramento da Floresta Amazônica Brasileira por Satélite) for the Brazilian Legal Amazon (Almeida et al., 2022).

The total GFED5 burned area for each 0.25° grid cell during the MODIS era (2001-2020) was estimated as the sum of the adjusted MODIS *normal* burned area aggregated over all vegetation types, and the additional burned area from cropland fires, peat fires, and deforestation fires (Eq. (2)).

$$BA_{GFED5}(x,t) = \sum_v BA_{GFED5-norm}(x,t,v) + BA_{GFED5-crop}(x,t) + BA_{GFED5-peat}(x,t) + BA_{GFED5-defo}(x,t) \qquad (2)$$

More information on how each of the terms in Eq. (2) were estimated is provided in section 2.4. We also derived linear regressions between GFED5 burned area (including contributions from all fire types) and active fire detections from the ATSR



or VIRS during years when these datasets overlap in time. We used coefficients from these regressions along with ATSR and VIRS active fire data from the pre-MODIS era to extrapolate the GFED5 burned area over the 1997-2000 period (Section 2.5).

**2.2 Primary datasets for burned area development**

We used several datasets at the MODIS spatial resolution (500-m) for the purpose of global burned area estimation and adjustment. A list of the datasets used in this study is summarized in Table 2 and described in detail below.

**2.2.1 MODIS 500-m burned area**

The MODIS instruments on NASA's Terra and Aqua satellites have provided global fire data for more than two decades. In
this study, we used the Terra and Aqua combined Collection 6 monthly Burned Area data product (MCD64A1) (Giglio et al., 2018) as the core basis for deriving the GFED5 burned area dataset during 2001-2020. The MODIS MCD64A1 burned area data also served as a spatial mask to separate the higher-resolution burned area images from Landsat or Sentinel-2 and MODIS active fire data into two classes (denoted as *in* or *out*). Landsat or Sentinel-2 burned area within (overlapping with) the MCD64A1 pixels were used to derive commission correction factors. Landsat or Sentinel-2 burned area data and MODIS
active fires outside of the perimeter of MCD64A1 pixels were used together to estimate the omission correction factors. MCD64A1 data are available from NASA and the University of Maryland fuoco sftp site (sftp://fuoco.geog.umd.edu). All data files are in Hierarchical Data Format (*hdf*) at MODIS sinusoidal projection, and contain data layers of burn date, burn data uncertainty, quality assurance, and other attributes.

**2.2.2 MODIS active fires**

In addition to the MODIS burned area, we also used the Terra and Aqua MODIS Collection 6 Thermal Anomalies/Fire locations 1-km data product (MCD14ML) (Giglio et al., 2016). The MCD14ML product records actively burning fires that are considerably smaller than those detected by the MODIS burned area algorithm (Giglio et al., 2006a). The monthly MCD14ML text files record multiple attributes associated with each active fire pixel detected by MODIS. Each record contains the center latitude and longitude location, the date and time of the detection, the satellite platform (Terra or Aqua), the along-scan sample
position, the hot spot type, confidence level, and other attribute fields. The spatial resolution of the MODIS active fire product varies from 1 km × 1 km at nadir to 4.8 km × 2.0 km at the scan edge. We only used active fire pixels with fire type marked as *'presumed vegetation fire'*. In the calculation of non-crop type burned areas, we also filtered out pixels with confidence level smaller than 30% and scan angles greater than 0.5 radians (30°) (corresponding to a maximum pixel area of ~1.7 km$^2$) to reduce the geolocation error.

MODIS MCD14ML fire location data was used in this study for deriving scalars and estimating burned area (for all types of burning as shown in Eq. (2)). We also used the MCD14ML data to calculate annual fire persistence (Chen et al., 2013), which in turn was used to estimate the deforestation mask and derive burned area for fires associated with deforestation (see section





2.2.6 for more information). For the ease of comparison with other MODIS data, we converted the original active fire location data to monthly 500-m rasterized images of active fire area masks (AF$_{MODIS}$). In order to do this, we first determined the 500-m sinusoidal grid cell that contains each valid active fire detection. We then marked this grid cell and the 8 surrounding cells as the active fire area, taking into account that about 10% of the MODIS active fires were outside but within 1500 m of the boundary of the burned area patch (Hantson et al., 2013). While this assumption may not have a substantial impact on the total burned area estimate (as we used higher resolution burned area to calibrate the scaling coefficients), it does not translate into a constant distance buffer across all tiles of the sinusoidal grid. According to the source of the satellite, we created three active fire datasets (1 for Terra only, 1 for Aqua only, and 1 for both Terra and Aqua) for each month.

The orbital dynamics for Terra and Aqua satellites provide denser spatial coverage toward the poles (and more daily overpasses), which changes the absolute number of active fires that are detected for a fire of the same size, intensity, and duration (Giglio et al., 2006b). Thus, MODIS active fire observations have systematic differences in their fire detection efficiency over different latitudinal bands. This becomes an issue when we derive scalars for use in a single GFED region that spans a wide range of latitudes. To correct for this bias, we adjusted the MODIS derived active fire area by multiplying it with a latitude-dependent overpass coefficient (AF$_{MODIS,adj}$=AF$_{MODIS,ori}\times$cos(latitude)). After the adjustment with this unitless coefficient, the AF$_{MODIS}$ data are normalized to the overpass frequency at the equator. Note that this adjustment was performed during the process of deriving the omission error correction scalar, and also during the GFED5 burned area estimation by converting MODIS active fire data to burned area.

**2.2.3 Land cover type**

In this study we used the Collection 6 MODIS land cover type product (MCD12Q1) that was derived using supervised classification of MODIS reflectance data (Friedl et al., 2010). The MCD12Q1 product maps annual global land cover at 500-m resolution using multiple land cover classification schemes. We reclassified the IGBP classification in MCD12Q1 to include 20 land cover types (Table S1 of van Wees et al., 2022) with additional masks coming from the FAO Global Ecological Zones (GEZ) 2010 update (FAO, 2012) for separating the tropical, temperate, and boreal zones. The reclassified 500-m land cover type (LCT) files were saved in GeoTIFF format at MODIS sinusoidal projection and used in this study to assign burned area and active fire pixels according to normal vegetation burning types. Active fires sampled in pixels of classes 12, 14, and 19 (Table S1) were aggregated and used with ancillary information to estimate cropland burned area. As described below, areas in peatlands or undergoing active deforestation were separately identified and used to replace the modified MODIS land cover types shown in Table S1.

**2.2.4 MODIS 250m vegetation cover data**

For a given land cover type, there can be significant variations in the density of vegetation that influences the relationship between active fire detection and burned area and the efficacy of burned area detection. Here we used the annual Fractional



Tree Cover (FTC) layer in the 250-m Terra and Aqua combined Collection 6 Vegetation Continuous Fields product (MOD44B)
(DiMiceli et al., 2021; Hansen et al., 2005) to further aggregate burned pixels into FTC ($tc$) bins (a total of 10 bins, each with a width of 10%) for the purpose of estimating commission and omission corrections. The original data from USGS Land Processes Distributed Active Archive Center (LP DAAC, https://lpdaac.usgs.gov/products/mod44bv006/) were resampled to 500-m resolution at the MODIS sinusoidal projection to be compatible with other data.

**2.2.5 Tropical and subtropical peatland map**

The Global Wetlands Map for tropical and subtropical regions (60°S - 40°N) was produced by the Sustainable Wetlands Adaptation and Mitigation Program (SWAMP) using a hydro-geomorphological model based on an Expert System approach (Gumbricht et al., 2017). Here, we used version 2 of the Global Wetlands Map that is available for downloading at the Center for International Forestry Research (CIFOR) website (https://www2.cifor.org/global-wetlands/). Specifically, we combined the peatland cover layer in this dataset with MODIS data to extract the burned area and active fire area in tropical and
subtropical peatland areas. The original 231-m GeoTIFF data was reprojected and cropped to align with the 500-m MODIS sinusoidal grid.

**2.2.6 Fire persistence and deforestation mask**

MODIS active fire data were also used for deriving deforestation masks in this study. Fire persistence (FP), defined as the mean number of days with active fire detections within a region during a calendar year, has been shown to be a good indicator
of aggregated burning and often used for classifying deforestation fires (Chen et al., 2013; Giglio et al., 2006a; Morton et al., 2008). Here we counted the number of active pixels at 500-m resolution for each year using the MODIS active fire data (recorded by both Terra and Aqua), and then calculated mean FP values for each 5-km and 0.25° grid cell, respectively. The annual fire persistence data, together with the yearly MODIS FTC data (from MOD44B), were used to derive a 5-km annual mask for tropical deforestation. All active fire area ($AF_{MODIS}$) located within this mask were considered burning associated
with deforestation fires. The deforestation mask was created only for 0.25° grid cells within the tropical forest biome, with a mean fractional tree cover greater than 20%, and a mean fire persistence greater than 1.2. Within these cells, the 5-km sub-pixels with mean FP greater than 1.5 were used as the tropical deforestation mask. During 2001-2002, the FP value was derived using active fire detections recorded by Terra MODIS only. To adjust for the period with Terra-only data, we lowered the 0.25° FP threshold from 1.2 to 1.08 and the 5-km FP threshold from 1.5 to 1.23 in the calculation of the tropical deforestation
mask for 2001 and 2002. These new thresholds were optimized using the 2003-2020 period for which both Terra and Aqua data were available, so that the total deforestation area from Terra (summed over the tropics) was the same as that derived using both Terra and Aqua data.



## 2.3 Landsat and Sentinel-2 burned area datasets

In recent decades, Landsat or Sentinel-2 imagery has been used to generate burned area products at higher spatial resolution
in many regions for various purposes (Franquesa et al., 2020; Gaveau et al., 2021; Glushkov et al., 2021; Goodwin and Collett,
2014; Hall et al., 2020; Hawbaker et al., 2020; Roteta et al., 2019; Souza et al., 2020; Vetrita and Cochrane, 2019). We
gathered multiple sources of Landsat or Sentinel-2 burned area products (Table 1) for use in calibrating scaling coefficients
and validating our GFED5 burned area time series. Among these high-resolution products, some have been manually inspected
and quality assured, while others have been derived using an automated burned area algorithm.

### 2.3.1 Reference datasets from BARD


The Burned Area Reference Database (BARD) was created and compiled by the European Space Agency's (ESA) Climate
Change Initiative (CCI) program (Franquesa et al., 2020). This publicly available database includes several global and regional
burned area reference datasets from multiple international projects (Franquesa et al., 2020). These datasets were derived from
Landsat or Sentinel-2 imagery, have undergone internal quality checks including visual inspection, and have been reprojected
and reformatted with a uniform set of attributes and metadata descriptors. The BARD database is available for download from
the e-cienciaDatos repository (https://doi.org/10.21950/BBQQU7). The initial BARD database included 6 reference datasets,
with image tiles located in a continental or sub-continental region (NOFFi data in Greece, AFR and AFRS2 in Africa, and
CONUS data in the USA), or distributed across the globe (FireCCI Global 2008 and FireCCI Global 2003-2014). Another
global dataset (C3S) was added to BARD in 2021 (Lizundia-Loiola et al., 2021).

The BARD burned area perimeters are stored in shapefile format in UTM/WGS84 projection. Each dataset includes a summary
metadata *csv* file (which contains the acquisition dates of the pre-fire and post-fire images), a shapefile of all image locations
for each project, and shapefiles of all vectors within each image boundary that indicate the status of burning: (1) burned area,
(2) undetected area, and (3) unburned area.

### 2.3.2 Automated datasets

The reference datasets in BARD are quality-assured, but their spatiotemporal coverage is often limited. In addition to the
reference datasets in BARD, we used several regional or global burned area products that were derived from automated
algorithms developed for Landsat or Sentinel 2 imagery.

The National Burned Area Composite (NBAC) is a GIS database and system developed by the Canada Centre for Mapping
and Earth Observation and the Canadian Forest Service (Hall et al., 2020). By using the best data source(s) available for each
fire event, including field maps, airborne maps, 30-m Landsat imagery, and 1-km MODIS hotspots, this tool produced the area
of forest burned in Canada for each year since 1986. Note that burned area in cropland is excluded from this dataset. The
annual burned area data in shapefiles can be downloaded from the CWFIS Datamart (https://cwfis.cfs.nrcan.gc.ca/datamart).



An automated approach based on the time series of Landsat 5, 7, 8 and Sentinel-2A imagery was recently developed to classify burned area in the state of Queensland (QLD), Australia (Goodwin and Collett, 2014). This approach combines the spectral,
thermal, temporal and contextual information from satellite imagery. The monthly burned area data across QLD over 1987-2016 (Collett, 2022) are available for download at the Terrestrial Ecosystem Research Network (https://portal.tern.org.au/annual-scars-landsat-qld-coverage/22979). Each annual data file (in GeoTIFF format) contains the month of first detection for pixels at Landsat resolution (30-m).

The MAWAS dataset provides maps of annual burned area for part of the Mawas conservation program in Central Kalimantan,
Indonesia from 1997 through 2015 (no fires were recorded in 2008 and 2010) (Vetrita and Cochrane, 2019). Landsat imagery (TM, ETM+, OLI/TIR) at 30-m resolution serves as the primary imagery used for the burned area classification. A random forest classifier was used to separate burned and unburned 30-m pixels with inputs of composites of Landsat indices and thermal bands, based on the pre- and post-fire values. Annual burned area shapefiles are available for download at Oak Ridge National Laboratory Distributed Active Archive Center (https://doi.org/10.3334/ORNLDAAC/1708).

Drawing on time series analysis of Sentinel-2 imagery, Gaveau et al. (2021) created a new burned area dataset for 2019 in Indonesia (IDNS2) using a supervised random forest classification algorithm. An independent reference dataset (IDNS2Ref) was also developed by randomly sampling burned and unburned sites, and visually detecting a smoke plume, burn, or heat source from the archive of original Sentinel-2 images. Both the burned area dataset and the reference dataset are in shapefile format. The burned area dataset contains annual burned polygons across the region, while the reference dataset contains the
locations of reference points. Both datasets can be downloaded at the open access data repository Zenodo (https://doi.org/10.5281/zenodo.4551243).

An algorithm combining Sentinel-2 MSI images and MODIS active fire detections was developed and implemented by ESA CCI to generate the Small Fire Database which contains 20-m burned area estimations covering the Sub-Saharan region in Africa (Roteta et al., 2019). Version 1.1 product (FireCCISFD11) contains monthly burned area (in *GeoTIFF* format) is
available for 2016 at pixel or grid resolutions at the Centre for Environmental Data Analysis (CEDA) archive (https://catalogue.ceda.ac.uk/uuid/065f6040ef08485db989cbd89d536167). The updated version 2 product covers Sub-Saharan Africa for the year 2019.

MapBiomas is a multi-disciplinary network aiming to reconstruct annual land use and land cover information between 1985 and 2020 for Brazil (Souza et al., 2020). MapBiomas mapped fire scars in Brazil using mosaics of Landsat images and a
classification algorithm based on deep neural networks (Alencar et al., 2022). We downloaded the 2019 MapBiomas burned area product (MAPB) from the Mapbiomas website (https://mapbiomas.org/en/colecoes-mapbiomas-1).

Imagery from different generations of satellites in the Landsat program has been used by the United States Geological Survey (USGS) to generate annual burned area products in the contiguous United States (CONUS) (Hawbaker et al., 2020; Vanderhoof et al., 2017) for years since 1984. Here we used the Collection 1 CONUS Landsat Burned area dataset



(https://www.sciencebase.gov/catalog/item/587017d7e4b01a71ba0c5ff7) that was derived from an updated algorithm (Hawbaker et al., 2020) from the earlier Landsat Burned Area Essential Climate Variable (BAECV) algorithm. Note that burned area mapping using Landsat has been shown to be challenging in many agricultural and rangeland ecosystems (Vanderhoof et al., 2017).

Using Sentinel-2 optical satellite imagery, a spring-specific burned area dataset was created on the territory of the Russian
Federation from January 1 to May 15, 2020 (Glushkov et al., 2021). The mapping was based on a participatory crowdsourcing digitizing approach that was specifically designed for rapid land-change assessment. This burned area dataset (RUSS2) is available at the Greenpeace website in shapefile format (https://maps.greenpeace.org/maps/spring_fires_2020).

### 2.3.3 Geospatial data pre-processing

The higher resolution Landsat and Sentinel-2 burned area datasets (reference or automated) described above have different
data formats and resolutions. Before using them for adjusting MCD64A1 burned area, we processed these datasets and converted them into a uniform format. Specifically, for data in vector shapefile format, we first rasterized each reference image at 20-m resolution (close to the resolution of Sentinel-2 multispectral instrument MSI). The output GeoTIFF files have a sinusoidal projection, compatible with the sinusoidal grids used by the MODIS land data products. We then aggregated the 20-m images to 500-m resolution (still in sinusoidal projection) by calculating the total area of burned scars (BA), unburned
regions (UBA), and undetected regions (UDA) (if available in the original dataset) within each 500-m pixel. This preprocessing of the higher resolution burned area data enabled an easier spatial alignment with the MODIS data and facilitated the derivation of correction scalars used for adjusting MCD64A1 burned area described below.

### 2.4 Estimating burned area for the MODIS era (2001-2020)

#### 2.4.1 Deriving correction scalars for commission and omission errors

To estimate and account for commission and omission errors within the MCD64A1 burned area product (for the *normal* type burning), we calculated two sets of correction scalars ($\alpha$ and $\beta$ in Eq. (1)) using the area sums derived from resized Landsat or Sentinel-2 burned area and spatiotemporally aligned MODIS data at 500-m resolution. For each higher-resolution burned area image used for calibration, we extracted the MODIS burned area, active fire area, as well as land cover type data (including the derived masks for croplands, deforestation and tropical peatlands) and fractional tree cover data that are spatially
overlapped with the reference burned area (Figure 1). Note the same year data of vegetation and deforestation masks were used for pixel classification. If the detection date of a MODIS burned area pixel is within the temporal range defined by the pre-burning and post-burning dates of the reference image, this pixel will be marked as burned. We summed up the total burned area from MODIS MCD64A1 ($BA_{MCD64A1}$). Similarly, a 500-m resampled active fire grid cell is marked as burning if the recording date is within the temporal range of the Landsat or Sentinel-2 reference data. By using the MODIS image
corresponding to each reference tile, we defined two masks representing the area within ('*in*') and outside ('*out*') of the MODIS



burned scar. Note that all pixels marked as 'UDA' (undetected area) were explicitly excluded from both masks. With these masks, we summed up the total higher-resolution burned areas ($BA_{hr,in}$ and $BA_{hr,out}$) and MODIS active fires ($AF_{MODIS,in}$ and $AF_{MODIS,out}$) for each land cover type and each tree cover fraction bin (based on MODIS vegetation cover data). When calculating the $AF_{MODIS,out}$ sum, we excluded all pixels that were within the 500-m buffer of a MODIS burned pixel (i.e., the 8 345 pixels surrounding each burned pixel).

A look-up table of correction scalars for *normal* type burned area ($\alpha$ and $\beta$ in Eq. (1)) was derived using aggregation of these area sums from different reference data sets (Table 1). The commission scalar $\alpha$ was calculated as the ratio of total $BA_{hr,in}$ to total $BA_{MCD64A1}$. The omission scalar $\beta$ was defined as the ratio of total $BA_{hr,out}$ to total $AF_{MODIS,out}$. To enable the calculation for cases when either Terra or Aqua data are unavailable (see Sect.2.4.4 for detail), we created three sets of $\beta$ using $AF_{out}$ from 350 Terra only (T), from Aqua only (A), and from both Terra and Aqua MODIS (M). Burning pixels within the masks of cropland, tropical peatlands and deforestation were excluded from summation of areas and derivation of $\alpha$ and $\beta$ in Eq. (1).

Considering the balance between representativeness and sampling density soundness, we computed $\alpha$ and $\beta$ as a function of GFED region (14 in total, Figure 2), *normal* land cover type (17 in total, Table S1), and fractional tree cover bin (10 in total). For each GFED region, we selected several reference datasets for deriving correction scalars (Table S2) and left others for 355 independent validation of the adjusted burned area (Table 1). We preferentially used reference tiles with higher quality (in particular those from the BARD database), spatial coverage, and sampling for the calculation of scalars in each region. The reference scenes we selected spanned all major regions of vegetation fires (Figure 2) and were relatively evenly distributed across the globe. Since the Middle East (MIDE) and northern hemisphere South America (NHSA) regions did not have enough reference burned area samples, we merged the region MIDE with northern hemisphere Africa (NHAF) and the region NHSA 360 with southern hemisphere South America (SHSA), respectively (Table S2). For each region, we derived scalars $\alpha$ and $\beta$ as a function of land cover type and tree cover fraction bin using the area sums from all valid reference tiles. For each unique combination of FTC bin (*tc*) and land cover type (*v*) where total recorded data did not have sufficient samples (*i.e.*, $BA_{MCD14A1}$ or $AF_{MODIS,out} < 20$ km$^2$), we generated a global look-up table of the two scaling coefficients (also as a function of *v* and *tc*) using all available reference data (Figure 3) and filled in the gaps.

### 2.4.2 Deriving scalars for cropland, peatland, and deforestation fires

In this study, we separately estimated burned areas associated with three additional fire types other than the *normal* vegetation burning: fires used in croplands for agricultural purposes, fires in tropical and subtropical peatlands, and fires associated with deforestation. For these special burning types, large omission errors prevent the direct use of MCD64A1 burned area product. Instead, we derived correction scalars from reference dataset and estimated burned area by scaling the active fire data from 370 MODIS.





We used the GloCAB product to estimate the burned area associated with agricultural burning (Hall et al., submitted). In the GloCAB algorithm, crop type conversion factors ($\gamma_{crop}$) were calculated for several widespread burnable crops including winter wheat, spring wheat, maize, rice, sugarcane, and generic/other (Table S3). The crop type conversion factors were derived from manually mapped cropland burned area reference data regions spanning about 190,000 fields (about 72,000 km$^2$) over five

different countries: Russia, Ukraine, United States, Canada, and Brazil. These field-level, cropland-specific burned area reference data were created using a combination of all available 20-m Sentinel-2 Multi-Spectral Instrument (MSI), 30-m Landsat-8 Operational Land Imager (OLI), and 3-m PlanetScope imagery (www.planet.com), in conjunction with filtered VIIRS (VNP14IMGML) and MODIS (MCD14ML) active fire data. Assuming that each reference data region is associated with a majority crop type, conversion factors were calculated by weighting each burned field by its burned area fraction,

summing all burned fields within the region, and dividing by the total number of cropland-filtered, latitudinally-adjusted MODIS active fire detections within the spatial and temporal constraints of each reference area. Global crop type specific $\gamma_{crop}$ scalars were calculated based on the median values of a combination of reference regions (see Table S3 for the crop-type reference region combinations). We refer to Hall et al. (2021a) and Hall et al. (submitted) for more details about the cropland burned area reference regions and the GloCAB product.

For the peatland burning, we derived a scalar ($\gamma_{peat}$) based on the Landsat-derived burned area data (MAWAS) in Mawas, Indonesia. Using the MAWAS dataset, we tallied the total burned area during 2003-2015 (when Terra and Aqua data are also available) within the peatland mask. Note the MAWAS data contained unlabeled areas that were not mapped due to cloud obscuration and/or Landsat 7 ETM+ Scan Line Corrector gaps, and here we assumed these areas were unburned. We then summed up the total active fire area recorded by MODIS (Terra and Aqua combined) within the same study region during the

same years, and calculated the mean burned area associated with unit active fire area. Without the availability of data over global peatland, we assume the scalar derived from this high-quality regional data represents the mean burned area in peatlands associated with unit area of MODIS active fire.

Similarly, we used a regional dataset to derive a scalar ($\gamma_{def}$) that converts MODIS active fire area to burned area associated with tropical deforestation. By analyzing Landsat images, the National Institute for Space Research of Brazil (INPE) reported

yearly gross primary forest deforestation statistics through the PRODES project since 1988 (http://www.obt.inpe.br/OBT/assuntos/programas/amazonia/prodes). We separately summed the total area of deforestation and cumulative MODIS active fire areas within the deforestation mask in the Legal Amazon of Brazil during 2003-2020. While the deforestation and burned areas may not match on the annual scale since the burning of biomass can occur in multiple years following the deforestation or forest loss detection can be delayed (van Wees et al., 2021), we assume that over a long

period they should be roughly the same. The ratio of the cumulative area of deforestation to the total area of active fire was used as a scalar to convert monthly MODIS active fire areas to GFED5 burned area associated with deforestation in the tropics.





### 2.4.3 Using scalars to estimate GFED5 burned area during the MODIS area

For each month during the MODIS era (2001-2020), we partitioned 500-m burned pixels (BA$_{MODIS}$) into different categories based on vegetation type, tree cover fraction bin, and fire types. By counting the burned pixels located outside of the cropland

mask (with reclassified landcover type code equals to 12, 14, and 19), peatland mask, and the deforestation mask, we calculated the area sums for *normal* type vegetation burning ($BA_{MCD64A1}$) over separate unique combinations of vegetation type and tree cover fraction bin in each 0.25° grid cell. Similarly, we also summed up the total area of active fire pixels that are located outside of the monthly MODIS area ($AF_{MODIS,out}$) for each bin in each 0.25° grid cell. These monthly MODIS burned area and active fire data were then either combined with pre-calculated correction scalars for commission errors ($\alpha$, as described

above) or omission errors ($\beta$) to estimate the adjusted burned area for *normal* burning (Eq. (1)).

For each 0.25° grid cell, the cropland burned area was extracted from the GloCAB product. GloCAB calculates the effective cropland burned area by multiplying latitudinally-adjusted MODIS active fires (within the cropland mask) with crop type scalars ($\gamma_{crop}$; see Sect. 2.4.2). The GEO Global Agricultural Monitoring (GEOGLAM) Best Available Global Crop-Specific Maps (BACS) (Becker-Reshef et al., 2020; Whitcraft et al., 2019) were used for identifying winter wheat, spring wheat,

maize, and rice, and the 2010 Spatial Production Allocation Model (SPAM; https://www.mapspam.info/) was used for assigning sugarcane to individual MODIS cropland pixels. Any active fire occurring on a cropland pixel not classified as winter wheat, spring wheat, maize, rice, or sugarcane was assigned a "generic/other" scalar (Table S3) to estimate burned area. To account for any double burning within a 12-month period, the final monthly burned area is adjusted to ensure the 6-month cumulative sum (centered on the peak burning month) does not exceed the crop area within the 0.25° grid cell.

For fires associated with peat burning and deforestation burning, we calculated the monthly total area of active fires in each grid cell ($AF_{MODIS-peat}$ and $AF_{MODIS-defo}$). By multiplying the active fire data with the pre-calculated adjusting scalars ($\gamma_{peat}$ and $\gamma_{defo}$), we can estimate the total burned area associated with these two fire types.

To derive the final GFED5 burned area product, we aggregated the burned areas recorded in all fractional tree cover bins from *normal* burning, and added the burned area estimations from croplands, peatland and deforestation burning (Eq. (2)).

### 2.4.4 Estimating GFED5 burned area when MODIS data are incomplete or unavailable

Between 2003 and 2020, MODIS active fire data from both Terra and Aqua are available for the majority of the time period. As described above, we estimated the *normal* type burned area outside of the MODIS burned area (second term in the right-hand side of Eq. (1)) in each month during these years using the total AF$_{MODIS,out}$ (as well as the correction scalars for omission errors $\beta$) from both MODIS instruments on Terra and Aqua. Since the full-year Aqua data started from 2003, we used

AF$_{MODIS,out}$ derived from Terra MODIS only for the omission error adjustment in years 2001 and 2002. The omission error correction scalars ($\beta$) were also derived using Terra MODIS data only. Note the use of Terra-only data does not bring



systematic biases over large spatial scales since all sets of correction scalars were derived using the same higher resolution burned area as the reference target (Figure S2, Table S4).

Similarly, the scalars used for special fire types (cropland burning, peatland burning, and deforestation burning) were also
derived using the combined detections by Terra and Aqua MODIS for the calculation of burned area in 2003-2020. During 2001-2002 when the Aqua MODIS data were not available, we multiplied the Terra-only active fire data with region-specific coefficients ($\gamma_{cropT}$, $\gamma_{peatT}$, and $\gamma_{defoT}$) to estimate the monthly burned area for these fire types ($BA_{GFED5-crop}, BA_{GFED5-peat}, BA_{GFED5-defo}$). These coefficients were derived by combining previously-derived GFED5 burned area and active fire area recorded by the Terra MODIS for the period of 2003-2020. This adjustment ensures that there
was no significant bias between regional or global burned area sums after scaling using Terra-only data and using data from both Terra and Aqua (Figure S3). We derived a global map of the adjustment coefficient at 0.25° grid level. For grid cells with insufficient burned area during the Terra and Aqua period ($< 2$ km$^2$ yr$^{-1}$), we assigned the mean value derived using the area sums over the corresponding GFED region.

There were major science data outages for Terra (Jun 15 - Jul 3, 2001) and Aqua (Aug 16 - Sep 2, 2020) MODIS observations,
which may have resulted in substantial underestimation of active fire detections, and therefore the AF$_{MODIS,out}$ estimation in June 2001 and August 2020. To address this issue, during the second gap when only Terra data were available (August 2020), we used the Terra active fire data to calculate burned area outside of the MODIS burned area (using the same approach as for 2001-2002). During the first gap when neither Terra nor Aqua data were available, we first adjusted the Terra active fire data for June 2001 by assuming the active fire detection frequency during the gap remained the same as it was in the first half of
the month when data were available. The same Terra-only correction approach was then used to derive the AF$_{MODIS,out}$ for June 2001.

## 2.5 Estimating burned area for the pre-MODIS era (1997-2000)

During the pre-MODIS era (including November-December 2000 when Terra MODIS data were available but fire persistence was not derived), we used active fire data recorded by the Visible and Infrared Scanner (VIRS) or the Along Track Scanning
Radiometers (ATSR) to estimate GFED5 burned area. VIRS was aboard the Tropical Rainfall Measuring Mission (TRMM) satellite, and monthly active fire data products were produced for pan-tropical regions (38° S - 38° N) during 1998-2010 (Giglio et al., 2003). Nighttime ATSR (onboard the ERS-2) and AATSR (Advanced ATSR onboard the Envisat satellites) imagery have been used to generate monthly global active fire maps (World Fire Atlas) for June 1995 - March 2012 (Le Page et al., 2008). We resampled the VIRS active fire data and binned the ATSR active fire locations to 1° resolution (without an overpass
correction) to reflect the higher uncertainty and lower spatial resolution of available active fire datasets.

We used a 2-step approach similar to that described in van der Werf et al. (2017) to estimate monthly GFED5 burned area during 1997-2000. First, based on the common era (2001-2010 for VIRS and 2001-2011 for ATSR) when GFED5 BA data have been estimated from MODIS (Eq. (2)) and active fire data were available from VIRS/ATSR, we derived a linear





relationship between active fires and burned area sums over each GFED region (*reg*), for each dominated vegetation class (*vc*)
(Table S1), and for each of the 3 seasonal periods (*ss*; Early season: 1-6 months prior to the peak burning month; Middle
season: peak fire month; Late season: 1-5 months after the peak burning month). We forced the regressions with an intercept
of zero to reduce biases at low values. We assumed that in each region, the burned area from the regression (BA$_{regr}$) was more
reliable when the adjusted $r_{reg,vc,ss}{}^2$ value (representing the goodness-of-fit) was higher during the overlap period. When the
regional sums of VIRS/ATSR active fires and GFED5 BA had a low covariation (small $r_{reg,vc,ss}{}^2$) during the common era, we
instead relied more on the burned area climatology (BA$_{MODIS,clim}$) at each calendar month (*m*) to approximate the burned area
in the pre-MODIS era. The total regional sums of the burned area (aggregated over all vegetation classes), determined by
combining BA$_{regr}$ and BA$_{MODIS,clim}$ (Eq. (3)), were estimated for each monthly time step (*t*) during the pre-MODIS era (1997-
2000).

$$BA_{preMOD}(reg, t) = \sum_{vc}((r_{reg,vc,ss(t)}^2 \times BA_{regr}(reg, vc, t) + (1 - r_{reg,vc,ss(t)}^2) \times BA_{MODIS,clim}(reg, vc, m(t))) \qquad (3)$$

The preference of using VIRS or ATSR active fire data depends on the data availability. In tropical regions the VIRS-based
regressions were found to have better performance than the ATSR-based approach. Therefore, when both ATSR and VIRS
were available (1998-2000), we used VIRS data to estimate the burned area in Africa, Southeast Asia, Equatorial Asia, and
Australia. We used ATSR for other regions and when VIRS data were unavailable prior to 1998. If the ATSR or VIRS active
fires for any given month were outside of the seasonal dynamic range of active fires during the MODIS era, we instead used
linear regression derived from all of the monthly data during the MODIS era for that region.

In a second step, we distributed the monthly regional sum ($BA_{preMOD}(reg, t)$) to burned areas in each 1° grid cell based on
two spatial distribution functions (SDFs). These SDFs were normalized so that their individual sums are 1 within each GFED
region. $SDF_{AF}(reg, x, t)$ represents the spatial distribution of active fires detected by ATSR and VIRS during each month.
Since ATSR and VIRS have high omission errors for small fires, the distribution from this SDF was likely biased toward grid
cells where large fires occurred. In order to better capture the contributions from small fires, we also used a climatological
distribution $SDF_{BA,clim}(reg, x, m(t))$, which was derived from MODIS burned area averaged over 2003-2020. As shown in
Eq. (4), the weights of these two SDFs were determined by the spatial correlations between GFED5 burned area and
ATSR/VIRS active fires ($r_{reg,m(t)}$) (i.e., the performance level of the regression model based on ATSR/VIRS active fires
during the overlap period).

$$BA_{preMOD}(x, t) = BA_{preMOD}(reg, t) \times ((r_{reg,m(t)}^2 \times SDF_{AF}(reg, x, t) + (1 - r_{reg,m(t)}^2) \times SDF_{BA,clim}(reg, x, m(t))) \qquad (4)$$

Eq. (4) is similar to the pre-MODIS era burned area estimation algorithm described by van der Werf et al. (2017) but uses r$^2$
instead of r to partition burned area between the distribution of active fires and a MODIS-era climatology. It is important to
keep in mind that the ATSR and VIRS instruments were not specifically designed for fire detection, and they are less efficient
at detecting active fires than MODIS. Therefore, the quality of the GFED5 data during 1997-2000 is not on par with that in



the MODIS era (Figures S4 and S5). Specifically, the spatial variability of burned area (derived from ATSR and VIRS active fires) was not well resolved in many regions (Table S4). For the above reason, in the pre-MODIS era, we only reported the total GFED5 burned area over all land cover types (i.e., normal, cropland, peatland, and deforestation types combined) at a coarser resolution grid (1° × 1°).

We recommend that users carefully consider the discontinuities in the GFED5 data stream for their specific application.
Quantitative analysis of trends for a single grid cell or small group of grid cells is not advised prior to the MODIS era due to discontinuities in the density of active fires. While we have attempted to account for these discontinuities at a regional scale in building the cross-sensor time series, attribution of burned area to individual grid cells is affected by changes in the density of active fire observations.

## 3 Results

**3.1 Global total and spatial pattern**

During the MODIS era (2001-2020) when higher quality remote sensing data are available, the total GFED5 burned area over the globe is estimated to be 774 (multi-year mean) ± 63 (standard deviation of interannual variability) Mha per year (or 5.9±0.5% of global burnable area, defined as global land area not permanently covered by snow). The spatial distribution of burned area is highly heterogeneous, with high levels of burning occurring in the tropics and northern hemisphere temperate
regions (Figure 4). Among the 14 GFED regions, northern hemisphere Africa (NHAF; 242±25 Mha yr$^{-1}$) and southern hemisphere Africa (SHAF; 245±13 Mha yr$^{-1}$) together contribute to about 63% of global annual burned area (Table 3). Other tropical and southern hemisphere regions, including southeast Asia (SEAS; 59±8 Mha yr$^{-1}$), southern hemisphere South America (SHSA; 54±12 Mha yr$^{-1}$), Australia (AUST; 56±23 Mha yr$^{-1}$), central America (CEAM; 13±2 Mha yr$^{-1}$), northern hemisphere South America (NHSA; 9.4±1.7 Mha yr$^{-1}$), and equatorial Asia (EQAS; 3.7±2.0 Mha yr$^{-1}$) combined contribute an
additional 25% of burning each year. In the northern hemisphere, most burning occurs in central Asia (CEAS; 43±11 Mha yr$^{-1}$), boreal Asia (BOAS; 32±11 Mha yr$^{-1}$), and temperate North America (TENA; 6.4±0.9 Mha yr$^{-1}$), and these regions together account for 11% of global burned area. The other three temperate and boreal regions of the northern hemisphere, including Europe (EURO; 4.3±1.3 Mha yr$^{-1}$), boreal North America (BONA; 4.1±1.3 Mha yr$^{-1}$), and the Middle East (MIDE; 3.1±0.8 Mha yr$^{-1}$), contribute to the remaining 1% of annual global burned area.

Herbaceous plants and shrubs are the dominant vegetation types that burn globally, with about 60% of annual burned area occurring in open savanna (260±17 Mha yr$^{-1}$) and tropical grassland (203±22 Mha yr$^{-1}$) vegetation classes (Table 3). Around 81% of the burning in these ecosystems (374±34 Mha yr$^{-1}$) is located in Africa (Figure S6). Major forest fires occur around the perimeter of tropical rainforests and in many seasonally dry boreal and temperate forests. Tropical forests and woodlands account for 9% (71±6 Mha yr$^{-1}$) of global burned area, with the largest contributions occurring in southern hemisphere Africa
(SHAF; 26±2 Mha yr$^{-1}$), southeast Asia (SEAS; 22±4 Mha yr$^{-1}$), and northern hemisphere Africa (NHAF; 15±2 Mha yr$^{-1}$).



Burning in croplands (83±14 Mha yr$^{-1}$) is similar in magnitude to that in forests, with hotspots in central Asia (CEAS; 21±6 Mha yr$^{-1}$), northern hemisphere Africa (NHAF; 21±6 Mha yr$^{-1}$), and southeast Asia (SEAS; 13±2 Mha yr$^{-1}$). The global distribution of crop burning is broadly consistent with spatial patterns of agricultural land use and crop yield (Lobell et al., 2011). While the area burned associated with deforestation (3.8±1.2 Mha yr$^{-1}$) and peatland burning (2.7±0.5 Mha yr$^{-1}$)

accounts for only a small fraction (< 1%) of the global annual total, fuel consumption and the emissions of trace gases and aerosol particles associated with these fires can be substantial as a result of high fuel loading (Andela et al., 2022; van Leeuwen et al., 2014; Turetsky et al., 2015; van der Werf et al., 2017). Further, these fire types often are not balanced by regrowth or ecosystem recovery, and as a consequence they contribute significantly to land use carbon emissions and the build-up of atmospheric $CO_2$ in the atmosphere (Friedlingstein et al., 2022).

**3.2 Long-term trends and interannual variability**

From 2001-2020, global burned area declined by -1.21±0.66% yr$^{-1}$ (p < 2e7), corresponding to a cumulative loss of 24.2% or 187 Mha yr$^{-1}$ over the 20-year span (Table 4, Figure 5). Considering the longer but more uncertain 1997-2020 period, which draws upon VIRS and ATSR active fire observations for the pre-MODIS period, the decreasing trend from 1997-2020 is - 0.91±0.79% yr$^{-1}$ (Figure 5a), with a cumulative change over this interval of -21.8%. Our time series provides evidence that the

global decline in burned area reported by Andela et al. (2017) for the 1997-2015 period has continued for the more recent 2016-2020 period and is visible in both the GFED5 and underlying MODIS burned area time series. The annual declining trend during the MODIS-era is slightly weaker than that reported in Andela et al. (2017) when measured as a relative change, primarily because the adjustments here for omission errors allowed us to include many more small fires each year for which burned area seems to have declined less than for larger fires over the time period. This is illustrated in Figure 5a by the different

trends shown for MCD64A1 and GFED5 and in Figure S7 by a positive trend in the ratio of GFED5 relative to the underlying MODIS MCD64A1 burned area.

The main regions driving the global declining trend in burned area are located in tropical north Africa, northern Australia, southern hemisphere South America, and the Eurasian Steppe (Figures 6 and 7). Burned area in some boreal regions (e.g., eastern Siberia and western Canada) and temperate regions (e.g., croplands in China and India, forests in southern Australia)

shows an increasing trend during 2001-2020, but their contributions at a global scale are relatively small (Table 4). The northern boreal region north of 60°N has a non-significant positive trend of 2.5±3.0% yr$^{-1}$. In all other latitude bands, burned area has significant negative trends, with the most rapid decline (-2.7±2.1% yr$^{-1}$) occurring in northern hemisphere mid-latitudes (Figure 5).

By fire type, savanna (-1.0±0.8% yr$^{-1}$), grass (-1.7±0.9% yr$^{-1}$), and cropland (-2.5±1.6% yr$^{-1}$) fires have strong decreasing

trends, while forest (0.2±1.4% yr$^{-1}$), deforestation (0.5±5.4% yr$^{-1}$), and peatland (-0.1±3.1% yr$^{-1}$) fires do not show significant changes at a global scale during the past two decades. Even though the total amount of burned area in forests is considerably



lower than in savannas or grasslands, these biomes typically have high fuel loading, and therefore the increasing fire emissions in forested areas may partly offset the decreasing emissions from herbaceous and shrub burning (Zheng et al., 2021).

### 3.3 Characterization of seasonal variability

Seasonal patterns of burned area from GFED5 differ significantly in many regions compared to other burned area products (Figure 8). In many tropical areas, increases in burned area in GFED5 relative to other products are strongest at the end of the fire season. We also found that the relative change in burned area from MODIS MCD64A1 to GFED5 (defined as the enhancement ratio, $(BA_{GFED5} - BA_{MCD64A1})/BA_{MCD64A1}$) varies monthly and is typically smallest near the peak burning month and higher at the edges of the peak season (Figure 9). This pattern is consistent with the monthly variations of the active

fire area, which have a larger fraction located outside of MCD64A1 burned area near the beginning and end of the fire season (Figure S8). We also observe that the burned area in the late fire season occurs in areas with higher fractional tree cover (an important biophysical parameter describing the Earth's surface vegetation) (Figure S9). The mean fractional tree cover for burning pixels is generally higher after the month of peak burning, suggesting that more severe fires are likely to occur later in the dry season as dense vegetation cover with deeper roots takes longer to dry out. With the exception of boreal North

America (BONA), the active fire pixels outside of the burned area have slightly higher mean fractional tree cover than those located inside of the MCD64A1 burned area. The inclusion of burned areas associated with these 'out' fires with higher fractional tree cover may further modify the burned area seasonality.

  Some GFED regions, such as boreal Asia (BOAS), central Asia (CEAS), and temperate North America (TENA), have multiple peaks of burning, with one or more of the peaks associated with agricultural fires, which are generally small fires but tend to

occur in pulses during the pre-planting or post-harvest time periods. Large amplification of burned area for these small fires has been found to change the seasonal variations in the burned area (Figure S8). Such modification of seasonality likely improves the agreement between GFED5 and regional high-resolution products (see section 3.4 below).

### 3.4 Comparison with validation datasets from higher resolution burned area products

  We used regional burned area products derived from higher resolution satellite imagery (Landsat and Sentinel-2) to assess the

quality of the GFED5 burned area product derived in this study. As described above, these higher resolution products were intentionally excluded from our reference data list used for deriving scalars to correct for omission and commission errors.

  The comparisons have been performed in major fire regions across the globe (Figure 10). It is clear that, relative to the unadjusted MCD64A1 burned area, the GFED5 data generally has a significantly higher magnitude, although with variable enhancement ratios in different regions. The boost is particularly evident in tropical regions (including Africa, Indonesia, and

Brazil) where small fires and understory fires are frequent and fire mapping is difficult with coarse resolution sensors. For most regions, this enhancement leads to better agreement with the high resolution burned area data used for validation.





The GFED5 burned area also has good agreement with higher resolution data in terms of interannual variability (Figure S10) and regional spatial variability (Figure 11). Over the whole of Indonesia, the annual burned area in 2019 from MCD64A1 (2.2 Mha) is smaller than that derived from Sentinel-2 (3.1 Mha) (Gaveau et al., 2021). After adjustment, the 2019 GFED5 burned area in Indonesia increases to 3.8 Mha, and the bias has been substantially reduced in the interior tropical forests of Borneo and Sumatra (Figure 11).

Africa contributes to about 63% of the global burned area (Table 3). Recently, Ramo et al. (2021) and Roteta et al. (2019) published a new burned area product (FireCCI-SFD11) that was developed from 20-m Sentinel-2 imagery and covers the whole of sub-Saharan Africa for the year 2016. The total 2016 burned area in sub-Saharan Africa from GFED5 is 509 Mha after correction for commission and omission errors, which is considerably larger in magnitude than MCD64A1 (270 Mha). Compared to the SFD11 data, with the annual burned area of 489 Mha (Ramo et al., 2021), the GFED5 data reduces the overall bias from -45% to +3.8%. In addition, the non-uniform boost also increases the spatial correlation of the burned area to the SFD11 pattern ($r^2$ increases from 0.57 for MCD64A1 to 0.71 for GFED5 (Figure 11).

Annual MCD64A1 burned area in Queensland for the year 2016 is 7.0 Mha, which is 30% lower than the burned area sum derived from Landsat (9.9 Mha) (Goodwin and Collett, 2014). After adjustment for omission and commission errors, GFED5 burned area over Queensland in this year is 10.2 Mha, which is very close to the Landsat-derived estimate.

The total burned area in Brazil in 2019 as estimated by GFED5 is 31 Mha, which represents a 61% increase compared to the MCD64A1 burned area estimate. Relative to the Landsat-derived MapBiomas data (Alencar et al., 2022), the GFED5 burned area exhibits a positive bias in most regions of Brazil, with the exception of the central Amazon, Caatinga, and temperate Pampa region (Figure 11). GFED5 also has a significant increase in the number of grid cells (0.25° × 0.25°) with non-zero burned area compared to the MCD64 data (8630 vs. 5725). This number is closer to the number derived from Landsat data (9784).

Glushkov et al. (2021) used Sentinel-2 imagery to map burned area across the Russian Federation for the spring of 2020. Their product showed that there was five times more burned area compared to the MCD64A1 burned area estimate, likely due to Sentinel-2's higher sensitivity to small fires in croplands, grasslands, and abandoned croplands. Our approach, which ingests a cropland-focused burned area product, also resulted in a nearly 5-fold increases in the burned area for the spring of 2020 (Figure S11). The total burned area for the spring of 2020 (Jan - May) increased from 3.3 Mha as reported by MCD64A1 to 15.8 Mha. The GFED5 product is much closer in magnitude to the burned area derived from Sentinel-2 data (13.4 Mha) (Figure 10).

**3.5 Comparison with other global fire products**

Relative to the original MCD64A1 data, our adjustment for commission errors tends to exclude unburned islands and decrease the total burned area, while the adjustment to correct for omission errors adds information from undetected fires and increases



the burned area estimation. During the 20 years in the MODIS era (2001-2020), our estimation of global burned area (774 Mha yr$^{-1}$) is 93% higher than that from MCD64A1 (401 Mha yr$^{-1}$) (Table 5, Figure S12). Around 58% of the increase in mean

burned area (220 Mha yr$^{-1}$) is associated with burning in Africa. The basis for this amplification is tied closely to active fire detections that are outside of MCD64A1 fire perimeters, and for which higher resolution satellite imagery shows significant levels of burning. Active fire detections located outside of the MCD64A1 burned perimeter are significantly higher than those within the burned area perimeter in Southeast Asia, Equatorial Asia, Central Asia, Central America, and temperate North America (Figure S13). Accounting for burned area associated with these small fires increases the burned area estimates in

these regions substantially. The enhancement ratio from MCD64A1 to GFED5 burned area is also found to be related to the burning type and land cover type (Figure S14). The burned area in savanna and grassland has a relatively smaller amplification (less than 100%) than burning in forests and croplands.

In addition to the Collection 6 MCD64A1 data, we also compared the GFED5 burned area with historical versions of GFED (GFED4s) (van der Werf et al., 2017) and MCD64A1 (Collection 5) (Figure 12). There are four main differences between

GFED4s and GFED5. First, GFED4s was based on the Collection 5 MCD64A1 burned area data. The MCD64A1 burned area mapping algorithm was subsequently improved to offer better detection of small burns (Giglio et al., 2018). The resulting Collection 6 MCD64A1 burned area product detects more global burned area (26%) than Collection 5 data. Second, in the GFED4s algorithm, each active fire detection outside of the MCD64A1 burned area was assumed to be associated with a certain area of burning. This scaling factor, which is a function of region, season, and vegetation type, was estimated by

comparing the mean difference normalized burn ratio of active-fire pixels observed outside and inside the MCD64A1 burned perimeter (Randerson et al., 2012). While this approach helps capture some burned areas associated with small fires and boosted the global burned area estimates by about 40%, the GFED4s burned area was still likely conservative on the global scale considering the spectral signal for small fires is often hard to detect at MODIS resolution. Third, in addition to correcting the omission error that arises mainly from small fires, the GFED5 algorithm also explicitly adjusts the overestimation of

MCD64A1 burned area by excluding the unburned islands. Fourth, the cropland burned area methodology in GFED5 is completely new and based on a more comprehensive training data of burned area derived from high resolution Landsat, Sentinel-2, and PlanetScope imagery. By comparing GFED5 data with GFED4s, we find the GFED5 burned area is able to account for more small fires in many regions of the world (Figure 12). Relatively speaking, the long-term trend of global burned area from GFED5 is less negative than that from the GFED4s and MCD64A1 Collection 6 products (Table 5).

## 4 Discussion

### 4.1 Implication for reconciling bottom-up and top-down fire emissions estimates

Burned area is a key driver of the multiplicative approach proposed by Seiler and Crutzen (1980) to estimate emissions from biomass burning, which can then be used in atmospheric models to simulate the transport of trace gases and aerosols emitted





by fires. There are multiple lines of evidence that current global fire emission inventories may have underestimated fire
emissions, particularly particulate matter emissions, in many regions. Model-simulated pollutant concentrations are often
found to be lower than satellite and in-situ measurements in the fire-affected regions (Reddington et al., 2016). To match
observations of aerosol optical depth from surface or satellite data, fire emissions from inventories often have to be increased
by a factor of 2-3 (Johnston et al., 2012; Xu et al., 2021). There is also a substantial gap between bottom-up estimates of fire
emissions and those inversely derived from top-down approaches that assimilate trace gas and aerosol optical depth
observations (Ichoku and Ellison, 2014; van der Velde et al., 2021; Naus et al., 2022).

There are many factors that may contribute to the gap between bottom-up and top-down estimates of fire emissions, including
the calculation of aerosol optical depth (Reddington et al., 2019) as well as the underestimation of fuel combustion (van Wees
et al., 2022; Potter et al., 2022) and emission factors (Jayarathne et al., 2018; Stockwell et al., 2016; Wiggins et al., 2021;
Wooster et al., 2018). Our analysis and the development of the GFED5 burned area product suggest that the burned area data
in several widely used global emission inventories is substantially underestimated, largely due to the difficulty of detecting
and measuring burned area associated with small fires (Randerson et al., 2012). As a result, fire emissions and the fire impacts
on atmospheric composition may have been underestimated in past work that relied on global inventories, such as GFED4s,
in many regions. Our work suggests that an amplification may be expected in many other continental-scale regions. The
improved characterization of burned area in this study will contribute to a higher estimation of fire emissions in the next
generation of GFED.

We also find that the imbalanced increase in burned area during different stages of the fire season may help reduce the seasonal
cycle bias in modeled trace gasses such as CO. For example, top-down estimates of atmospheric CO using MOPITT
observations (van der Werf et al., 2006; Zheng et al., 2018) have identified a fire seasonal cycle in Africa that peaks later than
that from bottom-up approaches. Ramo et al. (2021) found that improved detection of small fires is likely to prolong the fire
season in Africa, as more small fires occur in the shoulder (beginning and ending months) fire season when the atmosphere is
more humid. Our analysis shows that in many regions, the relative increases in burned area from MCD64A1 to GFED5 are
higher in the early and late phases of the fire seasons than during the peak burning month (Figure 9a). Relatively more MODIS
active fires are detected outside of the MCD64A1 burned perimeter (Figure 9b) near the edges of the fire season, particularly
in the later fire season, and these fires occur over types with higher fractional tree cover. The inclusion of small fire contribution
in the GFED5 burned area product is therefore likely to reduce the seasonal variability of fire emissions, which is more
consistent with the top-down estimates.

**4.2 Implications for fire science and global carbon studies**

We have developed a multi-decadal time series of global burned area that includes the contributions from small fires and has
a better representation of cropland burning. The GFED5 burned area dataset is expected to contribute to fire science and global
carbon studies in several ways.



First, a longer and higher quality time series may improve fire prediction on subseasonal-to-seasonal (S2S) time scales. Machine learning techniques (Cohen et al., 2019) used in S2S fire forecasts require information input from multiple sources, such as the seasonality, long-term trends, recent fire observations, and external climate drivers (Anderson et al., 2022; Chen et al., 2020). The skill of the forecast depends heavily on the quality and duration of the training data. We expect that the

temporally consistent, higher quality burned time series from GFED5 can enhance the accuracy of S2S fire forecasting systems in many regions where higher-resolution time series from Landsat or government records are not available.

Second, the GFED5 burned area can be used to evaluate or improve the performance of prognostic fire models. Current fire models, such as those from the Fire Model Intercomparison Project (FireMIP), often use satellite observations to constrain the simulation. Therefore, the global burned areas from these models are shown to converge to the value representing historical

satellite data (Hantson et al., 2020). However, these models are often not able to adapt well to the changes in factors that influence the burned area. For example, they have been shown to be unable to reproduce the observed decline in global burned area (Andela et al., 2017) and may also underestimate the fire impacts on carbon cycle and vegetation distribution (Lasslop et al., 2020). An improved global burned area such as GFED5 may help calibrate the parameters used in these models and reduce the differences between observations and modeled results.

Third, the GFED5 burned area dataset may also be used to improve our understanding of the climate and human controls on the interannual variability of global fires. By tracking burned areas globally for four different burning types and 17 different non-crop land cover types (Tables 3 and 4), this new dataset reveals variable patterns and trends in burned area over different regions. The GFED5 time series of burned area confirms the decline of fire in many ancient grassland ecosystems that are under threat (Buisson et al., 2022). The new time series also indicates that burning in cropland regions has declined

considerably in many regions and at a global-scale. The economic and technological drivers behind the cropland trends require further study. At the same time, our analysis reveals a positive trend in the burned area in many high northern boreal ecosystems, including Siberia. These findings will help connect variability and long-term trends of fires to underlying climate, land use, and socio-economic data (Andela et al., 2017). An improved understanding of the factors controlling the burned area variability may also contribute to a better projection of future changes in fire risk in response to global environmental change

(Knorr et al., 2016).

Fourth, the global burned area data is an important piece of information for understanding the impact of fires on trends in atmospheric composition and radiative forcing. During the process of combustion, fires release large amounts of trace gasses and particulate matter into the atmosphere, changing the composition and air quality of the atmosphere and modifying the radiative balance of the Earth system. The GFED5 data is expected to help quantify the effect of smoke from fires on air quality

and mortalities rates, and validate trends in atmospheric chemical composition as well as the radiative forcing from prognostic fire models used within global earth system models (Li et al., 2019).



### 4.3 Uncertainties and limitations

Our study improves the quantification of long-term global burned area trends by reducing the biases associated with unburned islands and small fires. Specifically, compared to the earlier GFED4s version which used normalized burn ratio calculations (Randerson et al., 2012), the algorithm for adjusting omission errors has been considerably improved. However, there are still important limitations. These limitations stem from uncertainties in the input MODIS datasets for burned area and active fire time series and from the limited spatial extent (and accuracy) of higher-resolution reference burned area datasets we used to estimate corrections for commission and omission errors. Additional discontinuities are introduced when we attempted to harmonize the global time series across time periods when only ATSR, VIRS, or MODIS Terra observations were available.

Measurements by MODIS instruments aboard Terra and Aqua provide the basis for GFED5 burned area estimation during 2001-2020. Although the Collection 6 MODIS active fire detection algorithm is an improvement compared to Collection 5, the presence of omission errors (from fires obscured by thick smoke) and commission errors (false alarms in tropical ecosystems) in the MODIS active fire data (Giglio et al., 2016) may still contribute to uncertainties in the distribution of burned area estimation outside of the MCD64A1 burned area.

We used regional burned area products derived from Sentinel-2 or Landsat measurements to adjust the MCD64A1 burned area. These higher resolution products are able to map areas from small and fragmented burning, but they also have quality and representativity issues. In particular, the longer revisit time of these sensors may impair or impede the detection of fires in regions with frequent cloud cover and rapid vegetation regeneration (Hawbaker et al., 2017; Padilla et al., 2015). In addition, the spatial and temporal coverage of these high-resolution data is often limited. Therefore, spatial and temporal sampling designs that balance the quantity and quality of reference images are needed to make the best use of these data. In this study, we derive omission and commission scalars as a function of fire type, land cover type, and fractional tree cover, for each GFED region separately. In some of these aggregations the sample size is small, which may lead to large uncertainty in the derived scalars. Higher quality reference datasets in underrepresented regions such as Siberia, central America and northern hemisphere South America are especially needed for better calibration. In the current algorithm, we also assume that these scalars remain the same at different stages of the fire season, which ignores possible seasonal variations in fire size and severity as a consequence of changes in fire weather and fuel moisture.

During the pre-MODIS era, the GFED5 burned area data is estimated using active fire detections from ATSR or VIRS. This extrapolation does not introduce significant bias in many continental or sub-continental scales (Figure S4, Table S4). For GFED regions with the largest burned area (northern hemisphere Africa, southern hemisphere Africa, Southeast Asia, southern hemisphere South America, and Australia), the time series of regional burned area derived from ATSR or VIRS active fires generally have a high correlation ($r^2 > 0.89$) with the GFED5 time series derived from MODIS observations. The correlation is somewhat lower in other continental regions including temperate North America, Europe, and central Asia ($r^2$ between 0.66 - 0.86) likely due to the low sensitivity of ATSR to small fires in these regions. The spatial distribution of burned area is less



well resolved in many regions, particularly in boreal and temperate regions (Table S4). Therefore, we suggest using the pre-
MODIS GFED5 burned area data time series with caution, particularly with respect to spatial patterns and temporal variability
in small regions. Further analysis and integration of Advanced Very High Resolution Radiometer (AVHRR) (Giglio and Roy,
2020) and Landsat imagery and burned area product (e.g. Schroeder et al., 2016) may reduce uncertainties during this period.

The time series we developed here provides further confirmation that burned area is declining at a global scale. In the context
of interpreting the global cumulative loss of 24.2±13.2% during 2001-2020, and trends at finer levels of disaggregation, several
key sources of uncertainty require careful consideration. First, in many areas where fires are being lost from the landscape, the
largest fires are declining at a faster rate. This may be a consequence of landscape fragmentation (Andela et al., 2017), and
evidence for this is shown in Figure 5 where the coarser resolution MCD64A time series declines at a faster rate than GFED5.
As fires become smaller in size, they become more difficult to detect, which in turn introduces higher levels of uncertainty in
trend estimates. Second, as the record length grows, the cumulative effects of climate change may begin to influence satellite
fire detection and burned area algorithm performance. In this context, a key future step is to quantify trends in cloud cover and
how they may influence the efficacy of our retrieval algorithms. Third, as previously described, spatial discontinuities from
the harmonization of time series across periods with different sensor availability restrict our ability to estimate trends for
individual grid cells or small groups of cells. For this reason, in Figure 6 we only present a map of trends for the 2003-2020
period when consistent data from Aqua and Terra were active. Finally, we note that we use ordinary least squares regression
to estimate the slope and uncertainty of the slopes reported here; the assumption of a gaussian distribution of errors around the
fitted line is unlikely to apply in all locations.

### 4.4 Future directions

Terra and Aqua, two NASA platforms that provide MODIS data, have already passed their design lives and are expected to be
retired within the next few years. In order to continue to understand long-term fire trends and produce consistent global burned
area data after the MODIS era, we may need to combine the active fire data from other remote sensors such as VIIRS with
Landsat/Sentinel-2 datasets. Key challenges toward this goal include 1) building global wall-to-wall burned area maps from
20- and 30-m data and fusing them with earlier MODIS records, 2) creating new maps of global land cover classes and
continuous fields of tree/shrub/grass vegetation fractional cover, and fusing these products with earlier MODIS records, and
3) understanding the differences between VIIRS and MODIS active fire products and the implications for long-term trends. In
particular, the higher spatial resolution and better geolocation accuracy of VIIRS may lead to more accurate representation of
small fires over different land cover types.

In recent years, there have been efforts to aggregate pixel-based fire detections (active fire or burned area) into individual fire
objects in order to track the evolution and associated attributes of individual fire events (Andela et al., 2019; Artés et al., 2019;
Balch et al., 2020; Chen et al., 2022; Laurent et al., 2018). A future direction is to create a fire event dataset that tracks the
spatiotemporal evolution of burned area, radiative energy, and emissions associated with individual large fires. The emissions



estimation for individual fires can benefit from recent advances in fuel consumption quantification at 500-m spatial resolution (van Wees et al., 2022). Combining gridded GFED5 burned area data with event-based data is likely to provide a more comprehensive and detailed view of fire regime changes over the past few decades and potential future trajectories.

## 5 Data availability

The monthly GFED5 burned area data produced in this study are available in netCDF files. For years between 2001 and 2020, five layers of burned area (Norm: normal type, Crop: cropland burning, Defo: deforestation burning, Peat: peatland burning, Total: the sum of all burning) are provided at 0.25°×0.25° resolution. The 'Norm' layer contains burned areas in each grid cell (in $km^2$) separated by 17 major land cover types. For the pre-MODIS era (1997-2000), only the 'Total' burned area layer with reduced spatial resolution (1°×1°) is provided. We also provide two global maps of burnable area (with water and snow/ice
cover excluded) in each grid cell (0.25°×0.25° for the MODIS era and 1°×1° for the pre-MODIS era). The GFED5 burned area dataset is publicly available on the open repository zenodo (https://doi.org/10.5281/zenodo.7668423) (Chen et al., 2023). We will also deposit the GFED5 data on the GFED website (https://www.globalfiredata.org) and Oak Ridge National Laboratory Distributed Active Archive Center for Biogeochemical Dynamics (https://daac.ornl.gov). All data sources used to derive the GFED5 burned area are stated in the text and Tables 1 and 2.

## 6 Conclusions


In this study, we develop the fifth version of the Global Fire Emissions Database (GFED5) monthly burned area data spanning the 24-year period from 1997-2020 by fusing multiple coarse- and high-resolution data streams.

During the period of 2001-2020, we adjust the Collection 6 MODIS MCD64A1 burned area by correcting for both commission (unburned islands) and omission (incomplete detection of small fires) errors in all land cover types excluding croplands. The
commission error scaling factors are derived with spatiotemporally aligned burned area images from MODIS and from higher resolution Landsat or Sentinel-2 datasets. Omission errors are corrected by multiplying the MODIS active fire data with the Landsat or Sentinel-2 burned area data, both sampled outside of the MCD64A1 burned area. In addition, a cropland-specific burned area product was used to improve the representation of small fires within cropland. Accounting for both corrections, the derived GFED5 global burned area (774 Mha yr$^{-1}$) is 93% higher than the MCD64A1 burned area, with the largest
enhancements over crop and forest regions, and during non-peak burning months. Global burned area shows a decreasing trend of 1.21% per year over the 20-year span of MODIS era (2001-2020), and a smaller decreasing trend of 0.91% per year over 1997-2020.

Relative to MCD64A1 and other global burned area products, the GFED5 burned area has a better agreement with independent estimations from high resolution data, and may substantially reduce the disparity between the top-down and bottom-up
estimates of fire emissions. An important next step is to use these data with improvements in emissions modeling (van Wees

and van der Werf, 2019; van Wees et al., 2022) and emission factor observations (Wiggins et al., 2021) to generate a GFED5 emissions product to advance climate, atmospheric chemistry, and global carbon cycle studies.

**Author contribution**

YC and JTR conceptualized the paper. YC developed the algorithm, created and validated the data sets. JH developed the
algorithm and derived the burned area product for cropland burning. DVW developed the land cover type data. All co-authors contributed to the methodology improvements, and provided data resources used for deriving the GFED5 burned area. YC prepared the manuscript with contributions from all the co-authors.

**Competing interests**

The contact author has declared that none of the authors has any competing interests.

**Acknowledgements**

This work was supported by NASA's Modeling Analysis and Prediction program (80NSSC21K1362) and SERVIR Applied Sciences Team (80NSSC20K0590). Additional support was provided by NASA's Earth Information System (EIS-Fire) and the Netherlands Organisation for Scientific Research (NWO); Vici scheme research programme, no. 016.160.324. Support for J. Hall and L. Giglio was provided under NASA Carbon Monitoring System (CMS) grant 80NSSC18K0179.



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




**Tables**

**Table 1**. Multiple burned area datasets derived from Landsat (LS) or Sentinel 2 (S2) imagery that are used to calibrate scaling coefficients or validate the GFED5 burned area time series. The term 'Ref' refers to datasets with manual quality inspection such as those reported in the Burned Area Reference Database (BARD). The term 'Auto' refers to datasets created using an automatic approach based on machine learning.


| Name | Type | Satellite | Years | Region | Original data name | Reference | Data source | DOI |
|---|---|---|---|---|---|---|---|---|
| **Datasets used for calibration** | | | | | | | | |
| AFRS2 | Ref | S2 | 2016 | Sub-Saharan Africa | BARD - FireCCI Africa S2 | Franquesa et al., 2020 | e-scienceData | 10.21950/BBQQU7 |
| GLB0314 | Ref | LS | 2003-2014 | Global | BARD - FireCCI global 2003-2014 | Franquesa et al., 2020 | e-scienceData | 10.21950/BBQQU7 |
| GLB08 | Ref | LS | 2008 | Global | BARD - FireCCI global 2008 | Franquesa et al., 2020 | e-scienceData | 10.21950/BBQQU7 |
| C3S | Ref | LS | 2017-2019 | Global | BARD - C3S global | Franquesa et al., 2020 | e-scienceData | 10.21950/BBQQU7 |
| GREECE | Ref | LS | 2016-2018 | Greece | BARD - NOFFi | Franquesa et al., 2020 | e-scienceData | 10.21950/BBQQU7 |
| USA | Ref | LS | 2003, 2008, 2013 | CONUS | BARD - CONUS | Franquesa et al., 2020 | e-scienceData | 10.21950/BBQQU7 |
| NBAC | Auto | LS | 2018 | Canada | National Burned Area Composite | Hall et al, 2020 | CWFIS | 10.1071/WF19201 |
| IDNS2Ref | Ref | S2 | 2019 | Indonesia | Indonesia BA S2 | Gaveau et al., 2021 | Zenodo | 10.5281/zenodo.4551243 |
| MAWAS | Auto | LS | 2001-2015 | Mawas in Indonesia | Mawas Landsat | Vetrita and Cochrane. 2019 | ORNL DAAC | 10.3334/ORNLDAAC/1708 |
| **Datasets used for validation** | | | | | | | | |
| NBAC | Auto | LS | 2016 | Canada | National Burned Area Composite | Hall et al, 2020 | CWFIS | 10.1071/WF19201 |
| AFR | Ref | LS | 2016 | Sub-Saharan Africa | BARD - FireCCI Africa | Franquesa et al., 2020 | e-scienceData | 10.21950/BBQQU7 |
| QLD | Auto | LS and S2 | 2016 | Queensland in Australia | Queensland Landsat | Goodwin and Collett, 2014 | TERN | 10.1016/j.rse.2014.03.021 |
| SFD | Auto | S2 | 2016 | Sub-Saharan Africa | FireCCI SFD11 | Roteta et al., 2019 | CEDA | 10.5285/065f6040ef08485db989cbd89d536167 |
| USGS | Auto | LS | 2003-2018 | CONUS | CONUS BA Landsat | Hawbaker et al., 2020 | EarthExplorer | 10.5066/F7T151VX |
| MAPB | Auto | LS | 2019 | Brazil | Mapbiomas BA Landsat | Souza et al., 2020 | Mapbiomas | 10.3390/rs12060924 |
| RUSS2 | Auto | S2 | 2020 spring | Russia | Russia BA S2 | Glushkov et al., 2021 | Greenpeace | 10.1088/1748-9326/ac3287 |



| Name | Type | Satellite | Years | Region | Original data name | Reference | Data source | DOI |
|------|------|-----------|-------|--------|--------------------|-----------|-------------|-----|
| IDNS2 | Auto | S2 | 2019 | Indonesia | Indonesia BA S2 | Gaveau et al., 2021 | Zenodo | 10.5281/zenodo.4551243 |



**Table 2**. Primary coarse-resolution datasets used in this study to derive the GFED5 burned area.

| Name | Origin | Product/method | Res. | Purpose | Reference | Data source | DOI |
|---|---|---|---|---|---|---|---|
| MODIS burned area ($BA_{MCD64A1}$) | MODIS | C6 MCD64A1 | 500m | Derive 0.25° BA, for Separate $AF_{in}$ and $AF_{out}$ | Giglio et al., 2018 | Fuoco | 10.5067/MODIS/MCD64A1.006 |
| MODIS active fire ($AF_{MODIS}$) | MODIS | C6 MCD14ML | 1 km | Derive BA outside of $BA_{MCD64A1}$ | Giglio et al., 2016 | Fuoco | 10.5067/FIRMS/MODIS/MCD14ML |
| Land cover type (LCT) | MODIS | Reclassified C6 MCD12Q1 | 500m | Separate BA over LCT bins | Van Wees et al., 2022 | Van Wees et al., 2022 | 10.1111/gcb.15591 |
| Fractional tree cover (FTC) | MODIS | C6 MOD44B | 250m | Separate BA over FTC bins | Dimiceli et al., 2021 | LPDAAC | 10.5067/MODIS/MOD44B.006 |
| Peatland cover (PeatM) | CIFOR | Tropical and Subtropical Wetlands Distribution V2 | 231m | Separate peatland BA | Gumbricht, 2012 | CIFOR | 10.17528/CIFOR/DATA.00058 |
| VIRS active fire ($AF_{VIRS}$) | VIRS | VIRS AF | 0.5° | Estimate BA in pre-MODIS era | Giglio et al., 2003 | Fuoco | 10.1080/0143116031000070283 |
| ATSR active fire ($AF_{ATSR}$) | ATSR | World Fire Atlas Alg1 | 0.5° | Estimate BA in pre-MODIS era | Arino et al., 2012 | ESA DUE | - |
| Fire persistence (FP) | MODIS | Derived from $AF_{MODIS}$ | 5km | Derive deforestation mask | Randerson et al., 2012 | This study | - |
| Deforestation mask (DefoM) | MODIS | Derived from FP, LCT, FTC | 5km | Separate deforestation BA | This study | This study | - |
| Cropland burned area | Multiple satellite data | GloCAB | 0.25° | Used for cropland BA estimation | Hall et al. submitted | Hall et al. submitted | - |






**Table 3**. Annual mean GFED5 burned area (in Mha yr$^{-1}$) for different GFED regions, different burning types, and different land cover types (for *normal* burning) during 2001-2020.

| Region | Total | Normal land cover type | | | | | | | | | | | | | Crop | Peat | Defo |
|---|---|---|---|---|---|---|---|---|---|---|---|---|---|---|---|---|---|
| | | Tundra | Sparse boreal forest | Boreal forest | Temp. grass | Temp. shrub | Temp. mosaic | Temp. forest | Trop. grass | Trop. shrub | Open savanna | Woody savanna | Trop. forest | Other | | | |
| BONA | 4.12 | 0.16 | 0.97 | 1.40 | 0.01 | 0 | 0.12 | 0.06 | 0 | 0 | 0 | 0 | 0 | 0.04 | 1.37 | 0 | 0 |
| TENA | 6.37 | 0 | 0 | 0 | 1.92 | 0.10 | 0.76 | 0.27 | 0 | 0 | 0.06 | 0 | 0 | 0.13 | 3.09 | 0.02 | 0.01 |
| CEAM | 12.62 | 0 | 0 | 0 | 0.30 | 0.13 | 0.59 | 0.09 | 1.02 | 0.01 | 2.08 | 3.30 | 2.49 | 0.04 | 2.44 | 0.08 | 0.11 |
| NHSA | 9.44 | 0 | 0 | 0 | 0 | 0 | 0 | 0 | 3.38 | 0.01 | 3.99 | 1.01 | 0.42 | 0.03 | 0.36 | 0.22 | 0.06 |
| SHSA | 53.89 | 0 | 0 | 0 | 1.58 | 0.28 | 3 | 0.12 | 12.46 | 0.46 | 21.35 | 3.92 | 4.1 | 0.29 | 4.35 | 0.58 | 1.63 |
| EURO | 4.26 | 0.02 | 0.04 | 0.06 | 0.37 | 0.03 | 0.73 | 0.1 | 0 | 0 | 0 | 0 | 0 | 0.08 | 2.82 | 0 | 0 |
| MIDE | 3.14 | 0 | 0 | 0 | 0.26 | 0.08 | 0.05 | 0.02 | 0.03 | 0.03 | 0 | 0 | 0 | 0.06 | 2.60 | 0 | 0 |
| NHAF | 242.1 | 0 | 0 | 0 | 0 | 0 | 0 | 0 | 98.32 | 0.05 | 100.7 | 7.28 | 14.63 | 0.15 | 20.52 | 0.33 | 0.31 |
| SHAF | 244.5 | 0 | 0 | 0 | 1.51 | 0.12 | 0.23 | 0.06 | 58.02 | 4.35 | 116.9 | 33.09 | 26.4 | 0.29 | 2.47 | 0.56 | 0.77 |
| BOAS | 32.38 | 1.84 | 5.71 | 4.82 | 2.73 | 0 | 9.26 | 1.56 | 0 | 0 | 0 | 0 | 0 | 0.17 | 6.29 | 0 | 0 |
| CEAS | 42.85 | 0.09 | 0.21 | 0.15 | 17.34 | 0.03 | 2.18 | 0.51 | 0.02 | 0 | 0.11 | 0.16 | 0.14 | 0.46 | 21.39 | 0.02 | 0.04 |
| SEAS | 59.03 | 0 | 0 | 0 | 0.06 | 0.02 | 0.10 | 0.03 | 3.74 | 0.09 | 8.24 | 10.49 | 22.35 | 0.13 | 13.27 | 0.3 | 0.30 |
| EQAS | 3.71 | 0 | 0 | 0 | 0 | 0 | 0 | 0 | 0.06 | 0 | 0.69 | 0.82 | 0.52 | 0.03 | 0.64 | 0.56 | 0.39 |
| AUST | 55.54 | 0 | 0 | 0 | 0.88 | 3.83 | 0.76 | 0.38 | 25.7 | 16.28 | 5.6 | 0.06 | 0.07 | 0.03 | 1.80 | 0.03 | 0.14 |
| GLOBAL | 774.0 | 2.15 | 6.93 | 6.44 | 26.96 | 4.63 | 17.77 | 3.21 | 202.7 | 21.27 | 259.7 | 60.12 | 71.13 | 1.93 | 83.39 | 2.72 | 3.75 |






**Table 4**. Long-term relative trends of GFED5 burned area (% yr$^{-1}$) for different GFED regions, different burning types, and different land cover types during 2001-2020. The trends are calculated using the Ordinary Least Squares (OLS) regression method. Significant trends ($p < 0.05$) are highlighted in bold.

| Region | Total | Normal land cover type | | | | | | | | | | | | | Crop | Peat | Defo |
|---|---|---|---|---|---|---|---|---|---|---|---|---|---|---|---|---|---|
| | | Tundra | Sparse boreal forest | Boreal forest | Temp. grass | Temp. shrub | Temp. mosaic | Temp. forest | Trop. grass | Trop. shrub | Open savanna | Woody savanna | Trop. forest | Other | | | |
| BONA | -1.91 | **-3.19** | -1.16 | 0.28 | 2.45 | - | 5.23 | 4.92 | - | - | - | - | - | -1.41 | -2.5 | - | - |
| TENA | 0.19 | - | - | - | 1.57 | 0.61 | 0.93 | **2.84** | - | - | **-3.31** | - | - | -0.40 | -0.97 | -2.66 | 0.57 |
| CEAM | -0.68 | - | - | - | 1.64 | **4.01** | -0.56 | -0.27 | -0.47 | -1.40 | -1.22 | -1.11 | 0.08 | **1.99** | **-1.22** | -0.01 | 1.14 |
| NHSA | -0.89 | - | - | - | - | - | - | - | 0.49 | -0.96 | **-2.40** | -0.91 | 0.74 | 1.02 | -0.1 | -0.21 | 1.71 |
| SHSA | **-2.07** | - | - | - | **-4.44** | -4.88 | -1.71 | 0.94 | **-1.85** | **-4.84** | **-2.00** | -0.93 | -1.45 | 1.44 | **-3.62** | -0.28 | -3.27 |
| EURO | **-3.44** | -1.95 | -3.56 | **-3.82** | -2.12 | -0.30 | -0.95 | 0.23 | - | - | - | - | - | -0.04 | **-4.53** | - | - |
| MIDE | 1.49 | - | - | - | **2.91** | **3.85** | 2.14 | 2.96 | **3.64** | **4.39** | - | - | - | **2.09** | 1.19 | - | - |
| NHAF | **-1.23** | - | - | - | - | - | - | - | **-1.39** | 1.39 | **-1.16** | **2.42** | 0.63 | 1.04 | **-3.58** | 0.52 | **4.88** |
| SHAF | **-0.39** | - | - | - | **-1.77** | 0.59 | 0.74 | **3.37** | **-1.41** | -1.17 | 0.02 | **-0.72** | 0.60 | 1.45 | **-2.3** | 0.7 | **4.69** |
| BOAS | **-3.55** | **-6.62** | 1.00 | -0.99 | **-3.04** | -9.12 | **-4.88** | **-4.21** | - | - | - | - | - | -1.39 | **-6.93** | - | - |
| CEAS | **-3.28** | -5.97 | -0.76 | **-7.40** | **-3.95** | **3.89** | -1.92 | -1.19 | **-4.55** | -3.23 | **-4.77** | **-8.62** | **-6.71** | **5.51** | **-3.06** | 1.2 | **5.92** |
| SEAS | 0.56 | - | - | - | 1.77 | **5.47** | 2.55 | 2.83 | **0.91** | **2.50** | -0.03 | 0.65 | 0.15 | **3.85** | **1.33** | 0.71 | 3.42 |
| EQAS | -2.51 | - | - | - | - | - | - | - | 0.05 | - | -2.01 | -2.92 | -2.44 | 1.10 | -5.05 | -1.23 | -1.05 |
| AUST | -3.08 | - | - | - | -1.97 | -4.81 | 1.50 | 5.11 | **-2.74** | -4.47 | **-2.35** | -0.09 | 0.48 | -0.19 | 0.8 | 0.79 | 8.1 |
| GLOBAL | **-1.21** | **-6.20** | 0.62 | -0.90 | **-3.13** | -4.03 | **-2.96** | -1.15 | **-1.51** | -3.75 | **-0.71** | -0.19 | 0.29 | **2.05** | **-2.48** | -0.05 | 0.52 |




**Table 5**. Comparison of GFED5 mean annual burned area (Mha yr$^{-1}$) and long-term relative trends (% yr$^{-1}$, in parentheses) globally and in different GFED regions (See Figure 2 for locations) with other global burned area estimates. Significant trends ($p < 0.05$) are highlighted in bold.

ｌ100

| REGION | GFED5 (2001-2020) | MCD64A1 C6 (2001-2020) | GFED5 (1997-2020) | GFED5 (2001-2016) | GFED4s (2001-2016) |
|---|---|---|---|---|---|
| BONA | 4.12 (-0.91) | 2.23 (-0.45) | 4.13 (-0.80) | 4.11 ( 0.45) | 2.83 ( 1.72) |
| TENA | 6.37 ( 0.19) | 2.80 ( **2.60**) | 6.52 ( 0.51) | 5.95 ( 0.17) | 2.86 ( 2.01) |
| CEAM | 12.62 (-0.68) | 2.68 (-0.59) | 13.40 (-1.12) | 11.89 (-1.26) | 2.97 (-0.85) |
| NHSA | 9.44 (-0.89) | 5.42 (-0.04) | 9.36 (-0.28) | 8.94 (**-2.46**) | 5.25 (-1.88) |
| SHSA | 53.90 (**-2.07**) | 28.85 (-1.22) | 53.59 (-1.05) | 52.51 (**-2.73**) | 25.97 (-2.00) |
| EURO | 4.26 (**-3.44**) | 0.91 (**-3.45**) | 4.42 (**-2.59**) | 4.26 (**-4.36**) | 1.19 (**-4.08**) |
| MIDE | 3.14 ( 1.49) | 1.00 ( 1.91) | 3.01 ( **2.10**) | 2.90 ( 2.47) | 1.27 ( 2.00) |
| NHAF | 242.1 (**-1.23**) | 123.6 (**-1.89**) | 253.3 (**-1.49**) | 236.7 (**-1.18**) | 151.4 (**-2.24**) |
| SHAF | 244.5 (**-0.39**) | 147.3 (**-0.61**) | 243.1 (-0.09) | 232.0 (-0.23) | 172.9 (-0.71) |
| BOAS | 32.38 (**-3.55**) | 9.27 (-0.86) | 31.70 (-1.65) | 33.04 (**-3.76**) | 9.34 (-2.00) |
| CEAS | 42.85 (**-3.28**) | 17.5 (**-4.26**) | 40.54 (**-1.50**) | 43.28 (**-3.81**) | 22.06 (**-3.35**) |
| SEAS | 59.03 ( 0.56) | 13.9 ( 1.30) | 59.26 ( 0.83) | 55.41 ( 1.07) | 14.88 ( 1.81) |
| EQAS | 3.71 (-2.51) | 1.43 (-2.20) | 4.45 (**-4.34**) | 3.95 (-0.71) | 2.05 (-1.11) |
| AUST | 55.54 (-3.08) | 44.52 (-3.42) | 58.45 (**-2.55**) | 55.53 (-3.77) | 50.43 (-3.48) |
| GLOBAL | 774.0 (**-1.21**) | 401.4 (**-1.43**) | 785.2 (**-0.91**) | 750.4 (**-1.27**) | 465.4 (**-1.64**) |





**Figures**

**Figure 1**. Example images used for adjusting burned area for *normal* fire type. The 20-m burned area data from the 2016
FireCCI Africa Sentinel-2 dataset (for tile T34NEP) is aggregated to a 500m resolution burned fraction image (BAhr). The
MODIS 500-m data (AF: active fire area; FTC: fractional tree cover (%); LCT: land cover type, BAmod: MCD64A1 burned
mask) is used to create images that are spatiotemporally aligned with the BAhr image. The BAhr and AF pixels are further
separated into those within (BAhr_in) and outside of (BAhr_out, AF_out) the MODIS burned perimeter.

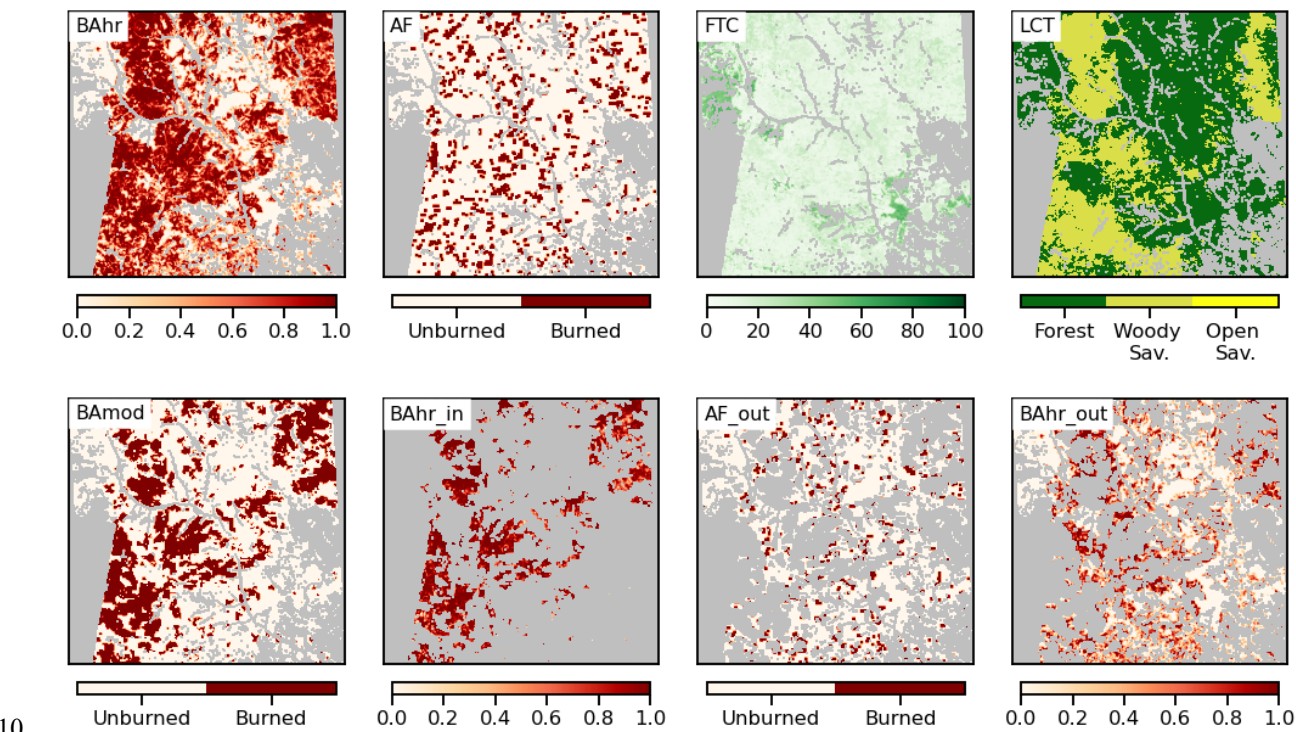






**Figure 2**. A global map of Landsat and Sentinel-2 reference burned area scenes used in this study to derive regional omission and commission scaling coefficients for *normal* fire type. Detailed information about the datasets can be found in Table 1 and Table S2. The inset shows the locations of 14 GFED regions.

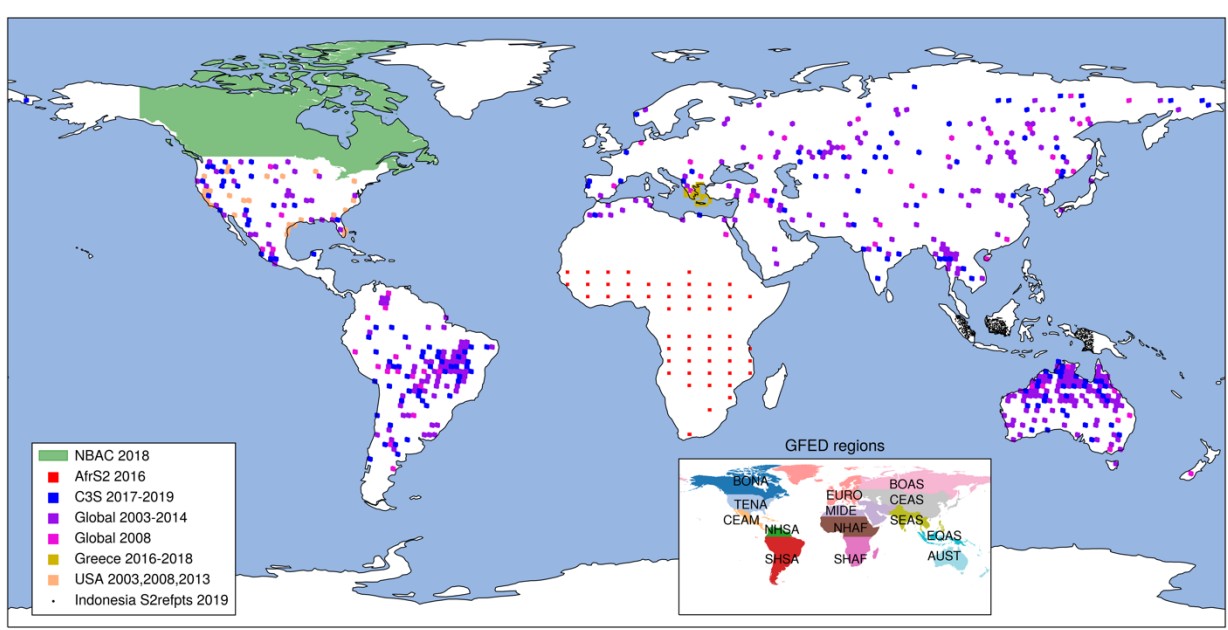

ι115



**Figure 3**. Mean values of commission scalars, omission scalars and burned areas for *normal* type fires in each bin combination

of fractional tree cover (FTC, in percent) and land cover type (LCT) were derived globally (shown here) and for each GFED

region (not shown) using reference burned area data as shown in Table 1 and Figure 2. Note the bins with small burned area

(BA$_{MCD64A1}$< 500 km$^2$ yr$^{-1}$, for commission scalar) or active fire area outside of the burned area (AF$_{MODIS,out}$<500 km$^2$ yr$^{-1}$, for

omission scalar) are shown in gray.

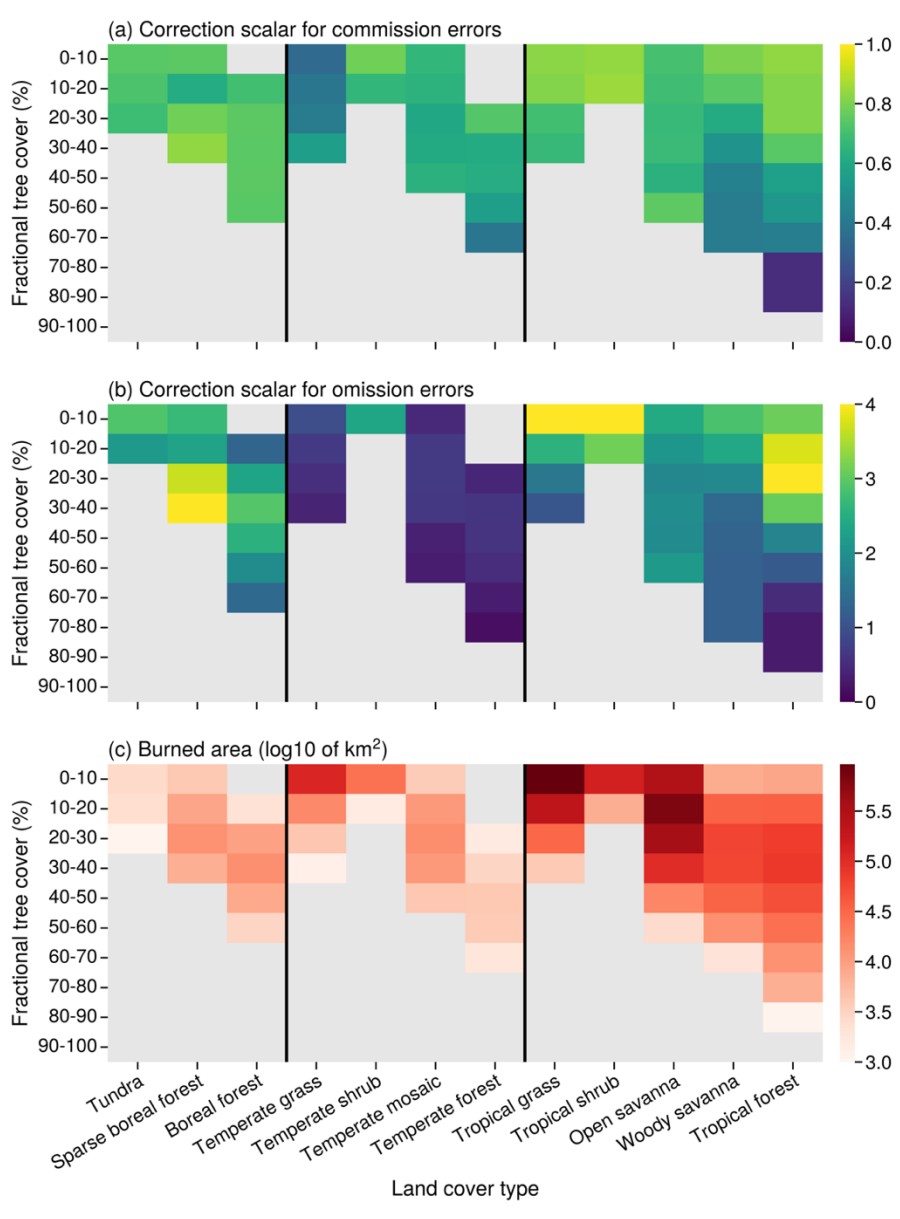


**Figure 4**. Global distribution of the mean annual GFED5 burned area, expressed as a percentage of the burnable land area in each 0.25° × 0.25° grid cell, from 2001 to 2020. The two area charts displayed above and to the right of the map provide a visual representation of the relative fractions of burned area along the longitude and latitude axes, respectively.

130

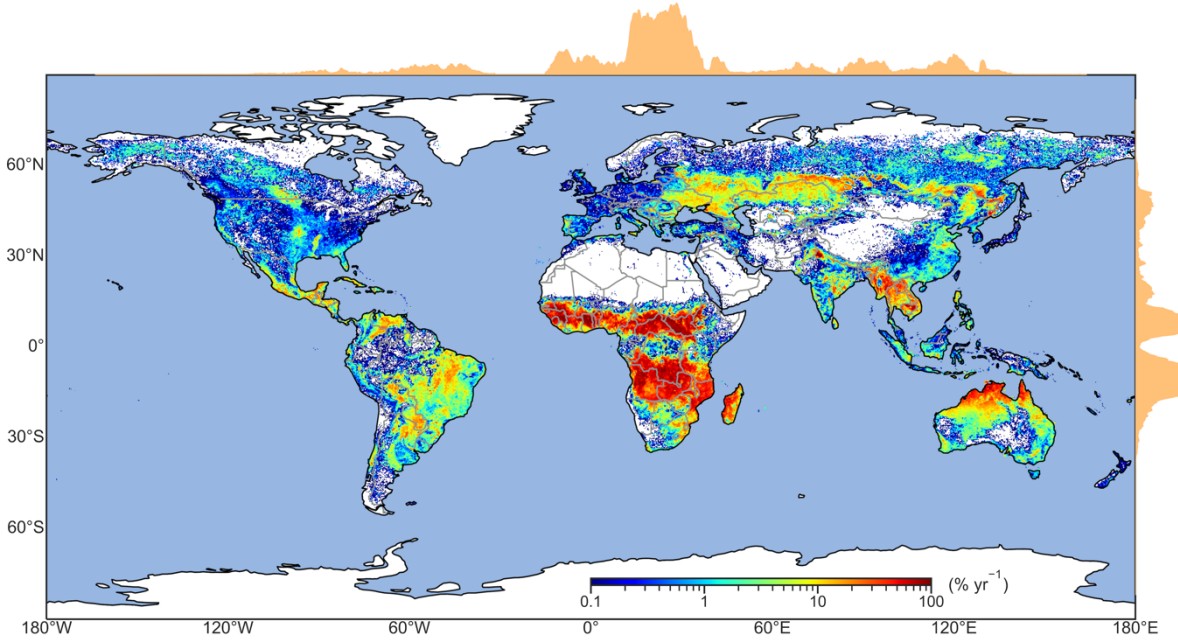



l135 **Figure 5**. Long-term trends in global burned area from GFED5 (in % yr$^{-1}$), normalized by the annual mean, for a) different time intervals and data products, b) normal type fires in aggregated vegetation classes, c) crop, peat and deforestation fires, and d) five latitudinal bands: Boreal (60°N-90°N), NHtemp (23.5°N-60°N), NHtrop (EQ-23.5°N), SHtrop (23.5°S-EQ), and SHextrop (90°S-23.5°S). Sign '*' before the number indicates a significant (p < 0.05) trend value. Labels and arrows on panels c & d denote the y-axis scale for each GFED5 burned area category.

l140

Earth System
Discussions
Science
Data

**Figure 6**. Global maps of GFED5 burned area trends during 2003-2020. (a) Map of the linear trend in units of absolute change in percent burned area for each 0.25° grid cell (% burned area $yr^{-2}$); (b) map of relative burned area trend in each grid cell (% $yr^{-1}$). The values in panel (b) were estimated by dividing the values in panel (a) by the all-year mean burned area shown in Figure 4.

1145

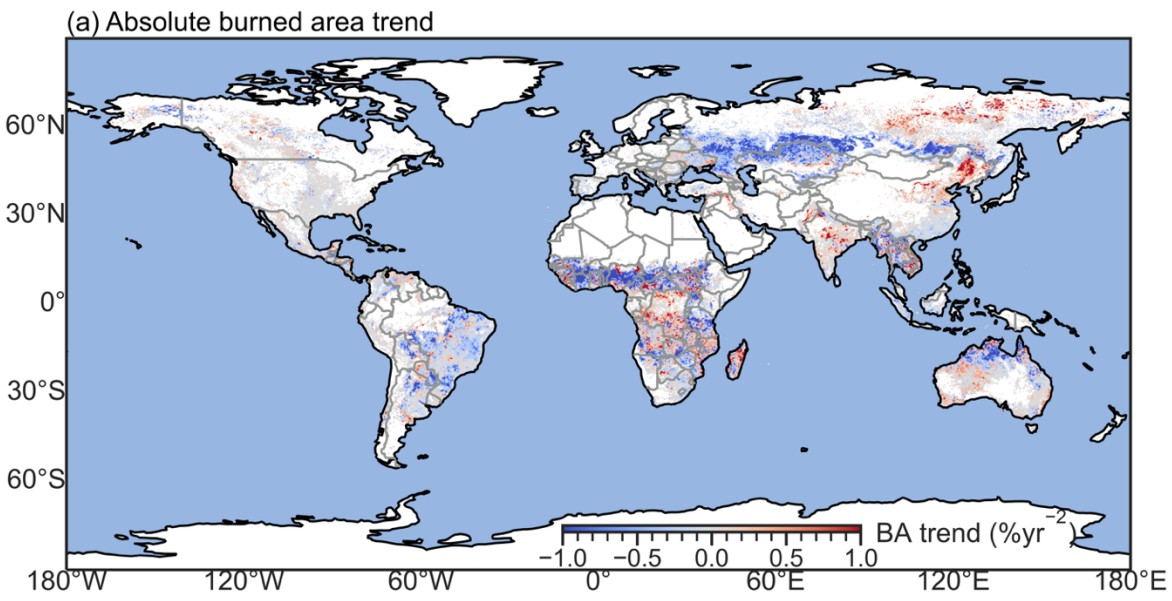

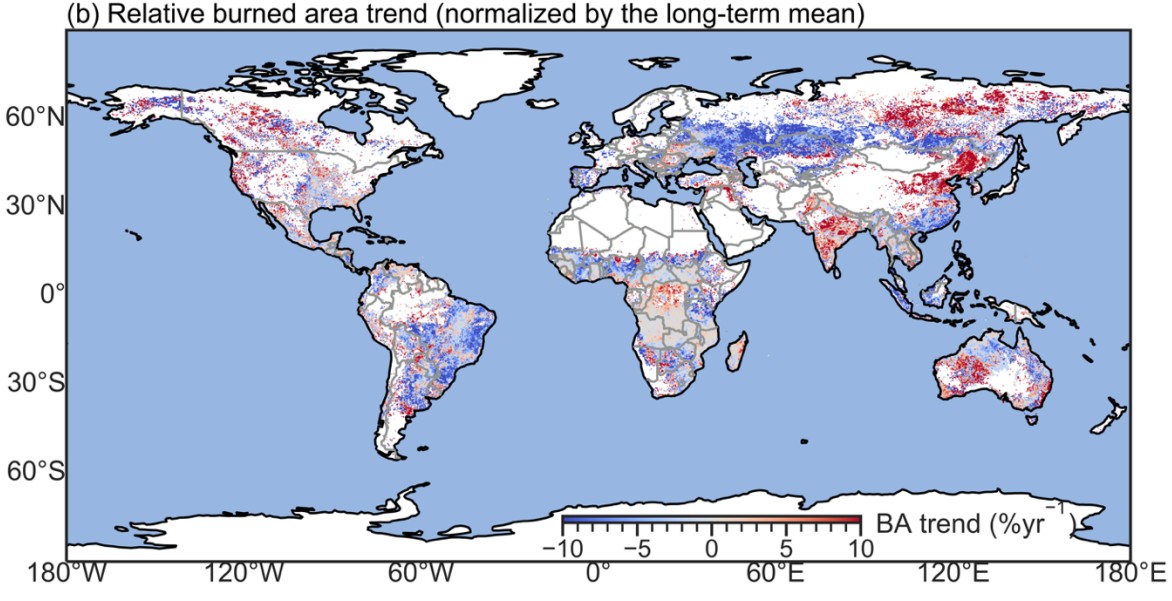



**Figure 7**. Time series of monthly (red lines) and annual (black lines) GFED5 burned area in 14 GFED regions from 1997 to
2020. The values in parentheses (in %yr⁻¹) are long-term trends during this period, normalized by the all-year mean values.
Sign '*' before the number indicates a significant (p < 0.05) trend value.



**Figure 8**. Comparison of the regional burned area seasonality from GFED5, MCD64A1, and GFED4s. The climatological monthly GFED5 and MCD64A1 burned area values are based on data from 2001 to 2020, and the GFED4s values are based on data from 2001 to 2016.


**Figure 9**. The ratios of (a) burned area from GFED5 to that from MODIS MCD64A1 ($BA_{GFED5}/BA_{MCD64A1}$), and (b) MODIS active fire area that is located out of the burned area to that located within the burned area ($AF_{out}/AF_{in}$). Each data point represents the mean monthly value, averaged in each GFED region over the 7-month period from 3 months before the peak burning month to 3 months after the peak burning month, for the 2001-2020 time period. Regions with multiple or no major fire seasons are not shown.

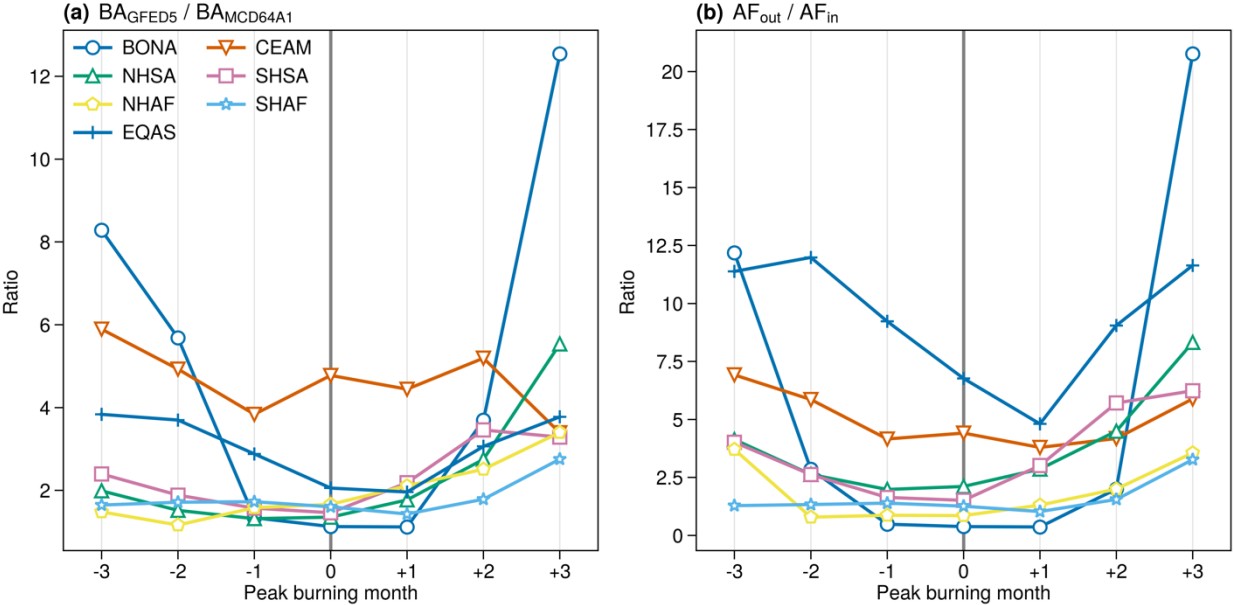



**Figure 10**. Evaluation of GFED5 burned area using independent products from higher-resolution imagery. For each source of Landsat or Sentinel-2 data (the datasets are listed in Table 1), the total burned areas (in Mha) from GFED5 and MCD64A1 are

calculated over the same region and periods as in the higher-resolution reference images. The regional burned area sums are compared in the inset bar charts, with the title indicating the region name, the year(s) of the measurements, and the name of the higher-resolution data source (in the parenthesis). In the comparison to Landsat data in CONUS and the NBAC data in Canada, we excluded the MCD64A1 and GFED5 burned area over croplands. We note that the GFED5 burned area and the higher-resolution datasets are not always directly comparable in this respect, as the burned area mapping from high-resolution

datasets may be incomplete due to persistent cloud cover or sensor failure.

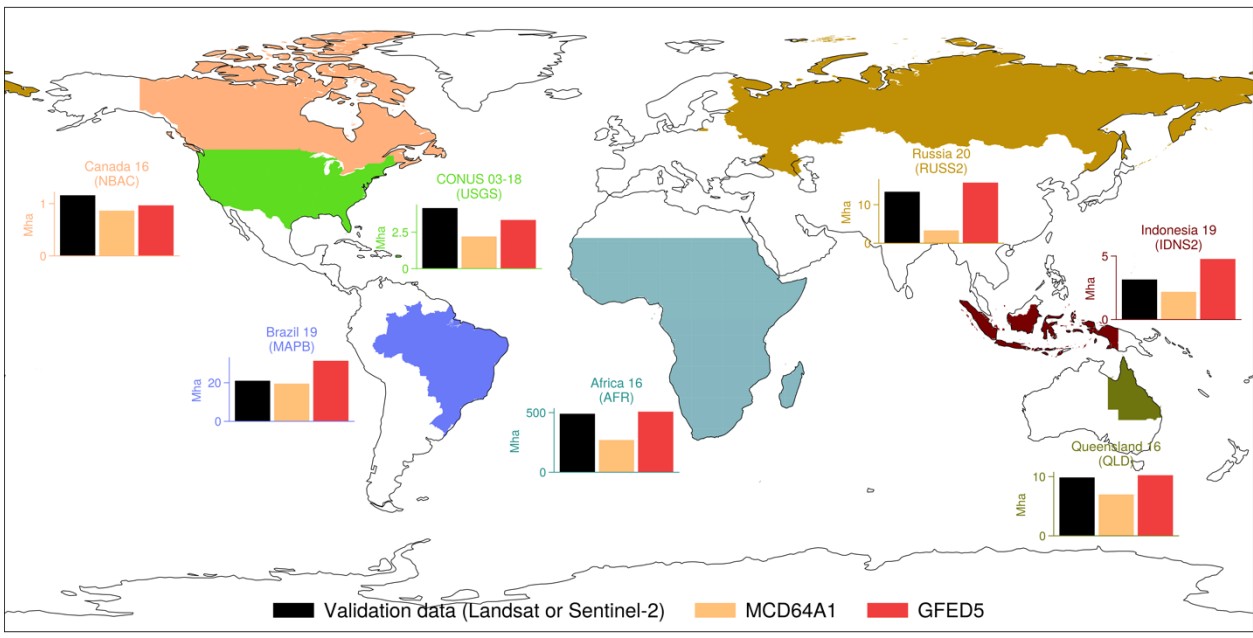




**Figure 11**. A spatial (0.25°×0.25°) comparison of burned area from GFED5 with burned area from MCD64A1 and higher
resolution satellites (SFD for Africa, MAPB for Brazil, IDNS2 for Indonesia; see Table 1 for details) is shown for (a) Africa,
(b) Brazil, (c) Indonesia.





**Figure 12**. Comparison of GFED5 burned area to GFED4s and different versions of the MODIS MCD64A1 burned area product (Collections 5 and 6) for the 14 GFED regions, based on the average burned area over 2001-2016.


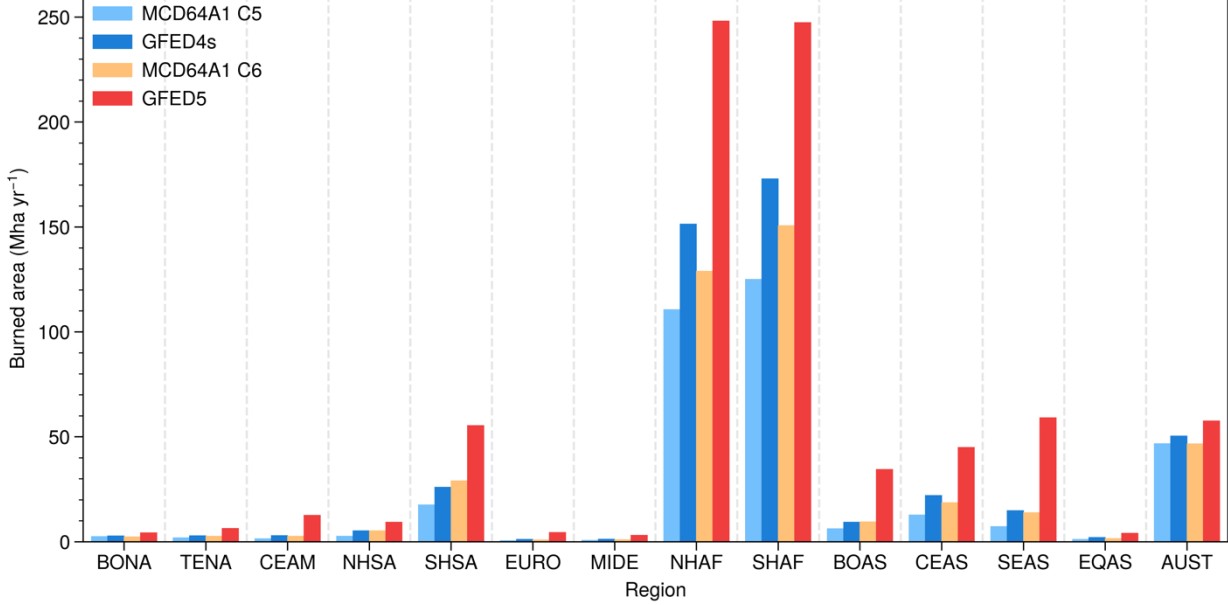