# Peer review of "Multi-decadal trends and variability in burned area from the 5th version of the Global Fire Emissions Database (GFED5)"

_Earth System Science Data, 2023_

## Author Comment (AC1)

**Responses to RC1**

**RC1:** 'Comment on essd-2023-182', Anonymous Referee #1
*>> We thank the reviewer for these helpful and constructive comments which are very helpful in revising and improving our paper. The point-by-point responses to these comments are listed below.*

This is a well-written review of a valuable data set and includes an interesting assessment of the data regarding spatial and temporal patterns of global fire emissions. The work also compares results to previously developed related datasets and reviews the value of this improved product for a variety of uses for air quality, climate studies, and more. The presentation is generally very good. I have only a few suggested technical edits and a the like, listed below.
>> Thank you for your positive feedback. We appreciate your comments and have carefully considered the suggested technical changes.

Data and Method (page 4)

The overview section is very important for this work because it serves as an opportunity to briefly describe the approach, so readers will not be obligated to read the full accounting of methods if it is not required for their purposes. A few minor edits will make this section even more helpful:
Line 113: This paragraph uses the word "higher" without reference to what it is "higher" than. I think it is higher than MODIS resolution, etc.. In general, when comparative words are used, there should be a clear connection to what they are referring to. I suggest a general re-wording to make this more clear.
>> Thanks for the suggestion on the usage of 'higher'. We have rephrased the first sentence to "***Our approach to estimate GFED5 global burned area … takes advantage of the high spatial resolution and detection sensitivity of burned area products from Landsat and Sentinel-2. Additionally, it leverages more frequent observations, global coverage, and extensive time series provided by the Terra and Aqua MODIS fire products, surpassing the temporal capabilities of Landsat and Sentinel-2.***" in the revised manuscript. We have also carefully evaluated all uses of 'higher-resolution' throughout the manuscript. In cases where there was no specific comparison target, we changed it to 'fine-resolution'.

Line 116: The 0.25 X 0.25 deg grid cell size of the dataset is mentioned here for the first time. The phrasing seems to assume that the reader knows that GFED products are of this spatial scale. I think it would be beneficial to be more explicit regarding the spatial scale and temporal time step in the opening paragraph of this section or in the introduction, if that seems more appropriate. As with the previous comment, revising the wording will remedy this issue.
>> Thanks for the suggestion. Although the resolution of the dataset has been mentioned in the introduction (Line 97 in the original version), we agree that it would be beneficial to explicitly state it again in the Data and Method section. In the revised version, we added '***at 0.25° spatial resolution and monthly time step***' in the first sentence of Section 2.1.

Line 129 (Eq. 1): The equation is given here out of order. Typically, equation variables are explicitly called out after the equation is introduced, rather than before and after, as done here. And, the left side equation variable is not explicitly defined until 2 paragraphs later (line 140). As a reader, it took some work going back and forth from prior to and after the equation to know the variables' meanings/definitions.

>> We have rearranged the positions of equations 1 and 2, and their descriptions. In the revised version, most of the explanations of the variables are located in the paragraph immediately following the equations.

Line 153: If I am correct and following this properly, you can reference back to Eq. 1 in this location to clarify the relationship between Eq 1 and Eq2. e.g: "…over all vegetation types (Eq. 1),…"

>> Thanks for the suggestion. In the revised version, we added the reference to Eq. 2 (note we switched the position of equation 1 and 2) in this sentence.

Line 155 (Eq. 2): Again, the Eq 2 left side term is not defined. You could reference it directly in the previous paragraph and call it out explicitly: "The total GFED5 burned area for each 0.25deg grid cell during the MODIS era (2001-2020; BAGFED5(x,t)) was estimated …" The subsequent sentence referencing section 2.4 is OK, but the reader is still left without an explicit definition of each term on the right side of the equation.

>> In the revised manuscript, the $BA_{GFED5}(x,t)$ term and all terms on the right side of the equation ($BA_{GFED5-norm}$, $BA_{GFED5-crop}$, $BA_{GFED5-peat}$, $BA_{GFED5-defo}$) are now defined immediately after Eq. 1 (note we switched the position of equation 1 and 2 in the revised version).

A revision of this short section will make this paper accessible to a wide audience for this important dataset.

Discussion (page 21):

Your discussion section is excellent. Informative, thought-provoking, and concise. I have two comments:

First: Line 646: In this section you use the terms "top down" and "bottom up". I strongly suggest this terminology be changed (I would like the community to stop using it), as it is confusing and not fully consistent.  For example, in this paper you use "top down" to mean estimation of aerosols based on atmospheric sensing retrievals from AOD (lines 654-655) and CO retrievals using MOPITT (lines 667-668). "Bottom up" is not explicitly defined, but I think the way it is written in the text equates "multiplicative approach of Seiler and Crutzen" (line 647) to "bottom up" as well as "satellite and in-situ measurements" (line 651), which are based on fuels and fuel combustion metrics found either via on-site measurements or fire energy methods (I am assuming). So, your definitions are fairly clear, with "bottom up" being an approach that measures components that drive emissions either directly or indirectly and "top down" being emissions estimated from atmospheric observations.

The issue is that other uses of the terms are divergent from this. Usually the "accounting" approach (Seiler & Crutzen, AKA "bottom up") is clear and consistent, with a variety of ways to come up with the amount of material that is combusted. However, some use "top down" in defining any use of remote sensing, including FRP-based combustion estimates e.g., Wiggins et al. 2021.  I have even seen GFED being referred to as a "top down" approach because it uses remote sensing for burn area estimation. To avoid this confusion, I suggest dropping the terminology and adopting what is used for general emissions estimation for sources other than biomass burning.

In emissions accounting outside of biomass burning, the community would use "activity-based methods" for "bottom up" accounting-style approaches. Your "top down" approaches would be called "atmospheric approaches". And any that combine these would be "hybrid approaches", which would include any data assimilation or modeling methods. zA recent NASEM report on GHG emissions accounting includes this generalized terminology  (see Page 3 of this report: https://nap.nationalacademies.org/catalog/26641/greenhouse-gas-emissions-information-for-decision-making-a-framework-going)

If you do choose to keep the terminology. Please work on the wording in this section to be clear on the definitions of top down and bottom up.
>> We agree that the use of 'top-down' and 'bottom-up' terminology is not well defined and often inconsistent in the literature. As suggested by the reviewer and the national academy report, we have changed the name of 'bottom-up' to 'activity-based' and 'top-down' to 'atmospheric-based' throughout the manuscript.

Second: Line 754: I am not sure what is meant by "cumulative effects of climate change" is it just changes in cloud cover? What other climate changes might make fire observations less now than before?
>> Here by referring to the "cumulative effects of climate change", we mean that the climate change may lead to modifications in atmospheric and surface conditions, which are likely causing systematic changes in accuracy, sensitivity, and reliability of fire detection from space. Climate-induced cloud cover change may be the most important one, but there are other possible factors such as the influence of changing canopy cover obscuring ground fires, the formation of dense  smoke plumes, and satellite orbit height and revisit time, etc. We have revised the relevant text in the revised manuscript as follows: *"Second, as the record length grows, climate change may systematically alter atmospheric and surface conditions, such as trends in cloud cover, and begin to affect the performance of satellite fire detection and burned area algorithms. In this context, a key future step is to quantify these changes and how they may influence the efficacy of our retrieval algorithms."*

Tables and Figures:

A few minor notes:

Table 4 and 5 show different numbers for Total burn area trend for BONA – a typo, I think. (-1.91 in Table 4; -0.91 on Table 5)
>> Indeed, there was a typo in Table 4: the long-term trend for BONA burned area should be -0.91 %/yr.  We have corrected this error in the revised manuscript.

I found Table 5 a bit confusing due to the order of the columns, but this is very minor. I would have put the full period GFED5 to the left (first column). The others are "paired" for a meaningful comparison, so maybe vertical lines that show the way to compare would help.
>> Thanks for the suggestion. In the revised version, we move the column of GFED5 (1997-2020) to the leftmost column and then separately have the comparisons for the periods of (2001-2020) and (2001-2016).

**Responses to RC2**

RC2: 'Comment on essd-2023-182', Johannes Kaiser
>> *We thank the reviewer for these helpful and constructive comments which are very helpful in revising and improving our paper. The point-by-point responses to these comments are listed below.*

GENERAL COMMENTS
The manuscript presents the updated burnt area estimates in the latest version of the well-established GFED inventory, GFED5. In order to fill the entire time series of 1997-2020 with monthly values and 0.25deg resolution for the entire globe, burnt area observations from the MODIS satellite instruments along with active fire observations from the MODIS, VIRS, and ATSR satellite instruments are being used. A new methodology is employed compared to earlier work: The omission and commission errors of the MODIS burnt area products with 500m spatial resolution are corrected based on a comparison to high-resolution (20-30m) observations of burnt area by the Landsat and Sentinel-2 satellites at several reference sites and time periods. Using a very large number of fitting parameters, the GFED5 burnt area time series is thus anchored to these high-resolution observations of burnt area.
>> Thanks for the good summary on our manuscript.

The data is highly relevant for the scientific community and the manuscript is overall well written and suitable for publication in ESSD. The authors have presented the context of burnt area estimation very thoroughly. However, the wider context of other research on vegetation fires and their emissions should be given a bit more comprehensively (and with more primary references); I recommend that the more senior co-authors edit the manuscript, in particular the Introduction and Discussion sections, in this respect.
>> Thanks for the suggestion. In the revised manuscript, we made several changes to widen the context and make it more tightly connected to previous studies. Key modifications are summarized below (see our responses to specific questions for more detail)

- In section 4.1, we added a full paragraph to review current emissions inventories from activity-based and atmospheric-based approaches. We cited more references accordingly.
- In section 4.2, we added more text on the implications of the current version of GFED relative to earlier versions of GFED.

Generally, many mathematical relationships are described in the text instead of formulas. In my opinion, the manuscript would be clearer, if more formulas were used.

>> In the revised manuscript, we re-organized equations 1 and 2, and modified descriptions of these equations to make them more clear. We also added a new equation (Eqn. 3) to mathematically describe the estimation of burned area associated with three special types: crop, peatland, and deforestation.

SPECIFIC COMMENTS

The authors fit very many parameters to generate the final burnt area product. The fact that this seems necessary has repercussions, e.g. on the interpretation of the MODIS burnt area product. To fully understand the implications of how much information is coming from which input, it is necessary to be more explicit about the number of fit parameters. My understanding is the following and I think a discussion (possibly corrected where I misunderstood) should be added to the manuscript:

>> Thanks for the suggestion. We have made several changes in the text and Table S2 to clarify the use of reference data and the number of fit parameters in the revised manuscript. Please check the responses below for detailed changes we made.

For normal land cover types (12) in each GFED region (14) and each tree cover bin (10), there are two parameters each. This yields 12 * 14 * 10 * 2 = 3360 parameters. Fig. 3 shows that about half of the tree cover bins have no data, so the number of parameters for deriving burnt areas from the MODIS product for the normal land cover types is closer to, say, 1700.

For periods with MODIS coverage by only one satellite, the set of above-listed parameters is fitted separately for Aqua and Terra. This yields another 2 * 1700 = 3400 parameters.

For six different crop types, global conversion factors are derived (6). For each of peatland burning and deforestation fires (2), a single scalar is derived and globally applied (2). These 6 + 2 = 8 parameters are used to derive the corresponding burnt area from MODIS active fire observations.

Judging by the vastly different numbers of parameters apparently required to derive realistic burnt areas from MODIS burnt area (1700 + 3400 = 5100) and MODIS active fire (8), admittedly for different fire types, it appears that MODIS active fire observations may contain more information on burnt areas than MODIS burnt area observations. I don't really believe this, but I think the authors need to discuss it to justify their use of MODIS burnt area observations.

>> We appreciate the reviewer's discussion on the number of parameters utilized in our method. Indeed, we have fitted a substantial number of free parameters to adjust the MODIS burned area and active fire data. In cases where there was sufficient reference (high resolution) data in a slot defined by a region, a particular land cover type, and a fractional tree cover bin, we derived a valid set of commission and omission scalars (see equation 2 in the revised

manuscript). For the core MODIS era when both Terra and Aqua data are available (2003-2020), the total number of valid scalars is 724, with half (362)  for adjusting omission errors and half for adjusting commission errors. Within the 362 commission (or omission) scalars, 225 were from the aggregations in individual GFED regions and 137 were from global fittings (Note we used the global parameters if there are less than 20 km2 of burning area from the reference data in a regional slot).

We kindly note that the MODIS burned area is the core base data for deriving the GFED5 burned area in 2001-2020. Based on the equation 2 (in the revised manuscript), the gridded MODIS burned area data were adjusted for commission errors, and the omission errors were adjusted using the MODIS active fire data outside of the MODIS burned perimeters. Same number of valid scalars (α and β, each with 12 land cover types, 14 GFED regions and global, and 10 tree cover bins) were used for adjusting MODIS burned area (alpha) and active fire data (beta). For non-normal types of burning (crop, peatland, deforestation), the quality of the MODIS burned area mapping is considered to be low. As a result we only relied on information from MODIS active fires when estimating the GFED5 burned area for these types. But overall, the MODIS burned area observations provided a substantial amount of information to the GFED5 burned area product.

For the pre-MODIS era, different scaling parameters of VIRS/ATSR active fires to burnt areas are used for each GFED region (14), each dominant vegetation class (16), and each seasonal period (3). Additionally, for each parameter, the goodness of fit is used to decide whether to use a climatology, doubling the number of parameters. This yields another 14 * 16 * 3 * 2 = 1344 parameters.

>> In the pre-MODIS era, when deriving the scaling parameters of VIRS/ATSR active fires, we aggregated the land cover type to 4 major vegetation classes (see Table S1). The total number of parameters is  about 14*4*3*2 = 336, smaller than that in the MODIS-era.

Additional parameters are described in lines 481-498. I struggle to understand this paragraph exactly and recommend rephrasing it.

>> This paragraph discusses how emission redistribution is executed spatially within each GFED region, utilizing two spatial distribution functions - one from GFED5 climatology, and the other from the monthly active fire distributions. We have modified this paragraph for clarity:

>*In a second step, we distributed the derived monthly regional sum of burned area to each 0.25° grid cell within a region. We assumed that the spatial distribution of monthly burned area within each GFED region area can be approximately represented by a combination of two spatial distribution functions (SDFs) (Eq. 5).*
>*<Eqn 5 here>*
>*The first spatial distribution is characterized by the number of active fires detected by ATSR and VIRS within a GFED region (reg) during each month (t). VIRS has a coarse spatial resolution (2 km at nadir), and ATSR can only detect fires at night. The approach of using ATSR or VIRS active fires may lead to a bias toward large fires which generally burn longer and emit higher radiative energy. Therefore, we also used a second spatial distribution function to better account for contributions from small, 'background' fires that were not detected by ATSR or*

*VIRS. This climatological SDF was derived from the GFED5 burned area (which contains more information from small fires than ATSR and VIRS) averaged over 2003-2020. The weights, representing the relative contributions of these two SDFs, were determined by the spatial correlations between GFED5 burned area and ATSR/VIRS active fires (i.e., the performance level of the regression model based on ATSR/VIRS active fires during the overlap period).*"

Overall, it seems necessary to use more than 5100 + 8 + 1344 = 6452 fit parameters to derive a realistic burnt area time series from the satellite products of burnt areas and active fires. Corrections seem particularly necessary and difficult for burnt area observations. This poses serious questions on the information content of the satellite observations, the danger of overfitting, and future methodologies for burnt area estimation, for example, whether machine learning and inclusion of further data sources might be the appropriate approach. The authors should discuss the implications of this for their product and future developments, including in Section 6.

>> As previously mentioned, we did indeed fit a large number of parameters to derive an adjusted estimate of the global burned area time series from satellite (mainly MODIS) products of burned area and active fires. Given the large variability in the errors of commission and omission in the MODIS product (see Figure R1 below), we derived these adjustment coefficients separately for different geographical regions and biomes. For each slot defined by GFED region, land cover type, and fractional tree cover, we computed these adjustment coefficients using high quality reference data from high-resolution burned area products. In cases where the total reference burned area within a slot is minimal (< 20 km2), we assumed that the slot-specific coefficient was not feasible, as an alternative, we used coefficients estimated using all reference data over the globe.

While we strived to utilize high quality reference data and allowed our algorithm to account for variations across different regions, land cover types, and fractional tree cover bins, we recognize that the adjustment coefficients in certain slots may have substantial  uncertainties, which stem from several sources: 1) a limited availability of reference data and 2) potential imperfections in the sampling of reference data. Addressing these uncertainties and enhancing the accuracy of global burned area estimation represent important future directions for our research. In response to this concern, we have incorporated the following text into the revised manuscript to outline the uncertainties in the current algorithm and our approach to mitigating these challenges.

In Section 4.3, we slightly modified the text to describe the parameter-related uncertainty in our approach:

*"In this study, we derived a significant number of omission and commission scalars, each tailored to specific fire types, land cover categories, and fractional tree cover levels, and calculated separately for each GFED region (Table S2). It is important to note that in certain cases, these aggregations were based on relatively limited sample sizes and imperfect spatial sampling approach, which can result in noteworthy uncertainties in the derived scalars. We particularly*

*highlight the need for improved reference datasets, especially in regions that are currently underrepresented, such as Siberia, central America, and the northern hemisphere of South America. Enhancing the quality and availability of reference data in these areas is of utmost importance for achieving more precise calibration in our methodology."*

In Section 4.4, we added a paragraph to discuss the future direction of gathering more fine-resolution data and the exploration of new machine learning algorithms:

*"Another pivotal avenue of research should involve the acquisition of supplementary reference burned area datasets generated from fine-resolution sensors. The inclusion of such datasets can significantly enhance the precision of adjustments applied to MODIS-based products. Additionally, it is essential to delve into enhanced techniques for mitigating spatial variability in both omission and commission errors. The accumulation of a substantial volume of high-quality reference data presents an opportunity to explore innovative machine learning approaches that can operate with fewer parameters, all while retaining the capability to perform cross-region adjustments. This endeavor holds the potential to alleviate issues related to over-fitting and ultimately augment the accuracy and applicability of wildfire burned area estimation methods."*

[Figure]

**Figure R1**. Mean values of commission scalars (left 2 columns) and omission scalars (right 2 columns) for selective normal type fires in each bin combination of fractional tree cover (FTC, in percent) and major land cover type (LCT) for each GFED region. In cases where the total reference burned area within a slot is minimal (< 20 km2), we assumed that the slot-specific coefficient was not feasible, as an alternative, we used coefficients estimated using all reference data over the globe (Figure 3 in the manuscript).

The discussion, in particular on emission estimates (Section 4.1), completely ignores well-established active fire-based inventories like FINN, GFAS, and QFED. This context is required here.

>> In Section 4.1, we have added a paragraph to review existing global fire emissions inventories (from both the activity-based and atmospheric-based approaches)

"*In recent decades, several global fire emission inventories have been formulated to help us better assess the impact of fire on climate and air quality, to understand how climate change affects the frequency and intensity of wildfires, and to develop better strategies to manage wildfires. Burned area is a key driver in the multiplicative approach proposed by Seiler and Crutzen (1980) for estimating emissions from biomass burning. Inventories derived using this activity-based approach include various versions of GFED (van der Werf et al., 2006; van der Werf et al., 2010; van der Werf et al., 2017), Fire Locating and Modeling of Burning Emissions (FLAMBE, Reid et al., 2009), and the Fire Inventory from NCAR (FINN, Wiedinmyer et al., 2011; Wiedinmyer et al., 2023), etc. Thermal energy measured by satellites at the top of the atmosphere was used in other global emissions inventories, such as the Global Fire Assimilation System (GFAS, Kaiser et al., 2012), the Quick Fire Emissions Dataset (QFED, Koster et al., 2015), and the Fire Energetics and Emissions Research (FEER, Ichoku and Ellison, 2014), drawing upon the linear relationship between the fire radiative energy and the burned mass (Wooster., 2002).*"

Concerning the statement on evidence for the underestimation of fire emissions (line 649 ff), I think this is overly simplified, since there are also cases (regions and chemical smoke constituents) that appear to be overestimated. For particulate matter, the question of underestimation depends heavily on whether "emissions" are considered at the top of the flame or many kilometers downwind of the fire since strong and quick formation of secondary organic aerosols happens in between. Therefore, experience, for example, in the development of the global aerosol model of the Copernicus Atmosphere Monitoring Service, has shown that much of the demonstrated discrepancy between bottom-up and top-down emission estimates can be attributed to unresolved processes in the model. Generally, the discussion of the discrepancies between bottom-up and top-down estimates must include a discussion of other sources of error than from burnt areas, i.e. from fuel load, combustion completeness, and emission factors.

>> We agree that the discrepancy between 'top-down' (now renamed to 'atmospheric-based') and 'bottom-up' (now renamed to 'activity-based') estimates of emissions are not necessarily due to underestimation in the 'bottom-up' approach, and may come from other sources. We have modified the text to clarify this.

"*Many factors can contribute to the gap between activity-based and atmospheric-based estimates of fire emissions. Some are due to underestimation of various processes, such as the fuel combustion (van Wees et al., 2022; Potter et al., 2022) and emission factors (Jayarathne et al., 2018; Stockwell et al., 2016; Wiggins et al., 2021; Wooster et al., 2 018). Others may be related to the model deficiencies in accurately resolving the*"

*physics and chemistry of fire pollutants in the atmosphere, such as the calculation of aerosol optical depth (Reddington et al., 2019)*"

The statements in Section 4.2 on Implications could, in my opinion, generally have been made for the earlier version GFED4s as well. It would be nice to show the added benefit of the new version; It's obviously more accurate, but what implications does that have?

>> We have added a paragraph in Section 4.2 for showing the added benefit of GFED5.

"*Fifth, in contrast to previous iterations of GFED, the GFED5 dataset incorporated fine-resolution data products and accounted for a broader spectrum of smaller-scale fires. In addition, we separately quantified burned areas in relation to typical landscape fires and to special fire activity within croplands, peatlands, and deforested regions. These enhancements provide GFED5 with superior attributes when compared to its predecessors, making it more valuable for many of the applications mentioned above, including S2S fire prediction, evaluating model performance across seasons, facilitating inversion-based studies, and improving the accuracy of long-term trend assessments.*"

TECHNICAL CORRECTIONS

According to my understanding of the English language, the word "section" should be capitalized when followed by a number, like "figure". This applies throughout the manuscript, e.g. lines 107-110.

>> We have replaced all 'section' to 'Section' in the revised manuscript.

l. 567: Delete the comment in brackets, or explain why the parameter is important.

>> As suggested, we have removed the text in brackets 'an important biophysical parameter describing the Earth's surface vegetation' from the revised manuscript.

l. 653: We have even found a factor of 3.4, i.e. larger than the cited range, cf. Kaiser et al. BG 2012.

>> We have added the reference of Kaiser et al., BG 2012 and changed the range from 2-3 to 2-4 in the revised manuscript.

Section 2.4.3 seems to be a repetition of previously given information and could be deleted.

>> Section 2.4.3 describes the detailed steps we combined the scalars (derived previously) with the MODIS burned area + active fires to actually derive the GFED5 burned area data. Although there are some overlaps between this section and the Method overview section 2.1, we respectively insist this is not simply a repetition of previously given information.

Fig. 8: If you plotted the x-axis on a log scale, it would be possible to see the differences in the shape of diurnal cycles as well as the magnitude of the emissions.

>> Thanks for the suggestion. We have replaced Fig. 8 with a new figure using log scale y-axis (see below)

[Figure]

Month

BA (Mha mo⁻¹)